# Uncovering diversity and metabolic spectrum of animals in dead zone sediments

Elias Broman [1,2,8✉], Stefano Bonaglia [1,3,8✉], Oleksandr Holovachov [4], Ugo Marzocchi [5,6], Per O.J. Hall[7] & Francisco J.A. Nascimento [1,2]

Ocean deoxygenation driven by global warming and eutrophication is a primary concern for marine life. Resistant animals may be present in dead zone sediments, however there is lack of information on their diversity and metabolism. Here we combined geochemistry, microscopy, and RNA-seq for estimating taxonomy and functionality of micrometazoans along an oxygen gradient in the largest dead zone in the world. Nematodes are metabolically active at oxygen concentrations below 1.8 μmol L$^{-1}$, and their diversity and community structure are different between low oxygen areas. This is likely due to toxic hydrogen sulfide and its potential to be oxidized by oxygen or nitrate. Zooplankton resting stages dominate the metazoan community, and these populations possibly use cytochrome c oxidase as an oxygen sensor to exit dormancy. Our study sheds light on mechanisms of animal adaptation to extreme environments. These biological resources can be essential for recolonization of dead zones when oxygen conditions improve.

[1] Department of Ecology, Environment and Plant Sciences, Stockholm University, Stockholm 106 91, Sweden. [2] Baltic Sea Centre, Stockholm University, Stockholm 106 91, Sweden. [3] Nordcee, Department of Biology, University of Southern Denmark, Odense 5230, Denmark. [4] Department of Zoology, Swedish Museum of Natural History, Stockholm 10405, Sweden. [5] Center for Electromicrobiology, Section for Microbiology, Department of Bioscience, Aarhus University, Aarhus, Denmark. [6] Department of Integrative Marine Ecology, Stazione Zoologica Anton Dohrn, Naples, Italy. [7] Department of Marine Sciences, University of Gothenburg, Box 461, Gothenburg 40530, Sweden. [8]These authors contributed equally: Elias Broman, Stefano Bonaglia. ✉email: elias.broman@su.se; stefano.bonaglia@su.se

The constant increase in global use of fertilizers and discharges of nitrogen (N) and phosphorus (P) is causing drastic changes to ocean biogeochemistry and increasing vulnerability of aquatic environments[1,2]. Nutrient-driven eutrophication is increasing not only along the coast, but also in otherwise nutrient-deficient open waters, fueling aquatic primary production worldwide[1]. Scarce water circulation and high rates of degradation can eventually lead to water column hypoxia ($\leq 63$ μmol $O_2$ $L^{-1}$ or $\leq 2$ mg $O_2$ $L^{-1}$) and anoxia (undetectable oxygen)[3]. This phenomenon, ocean deoxygenation, is further enhanced by global warming as higher water temperatures stimulate metabolic processes and decrease oxygen solubility[4]. Oceanic models anticipate a global decrease in the total oxygen inventory of up to 7% by 2100, with a number of oxygen minimum zones (OMZs) losing more than 4% oxygen per decade[5].

Anoxia in pelagic and benthic environments can be temporal and last minutes to hours as in the case of intertidal mud flats. Invertebrates can cope with these short-term events by activating anaerobic energy metabolism[6]. Anoxia, however, can persist for hundreds to thousands of years as in the case of certain stagnant bottom water of enclosed seas such as the Baltic and Black Seas[3,7]. In these systems, bottom water close to the seafloors is regularly characterized by very low oxygen ($\leq 22$ μmol $O_2$ $L^{-1}$), which precludes life to most animals[7]. These marine systems characterized by severe hypoxia or anoxia are often referred as dead zones[7]. While the term dead zone gives an idea of an ecosystem without life, it was shown that the core of large oceanic OMZs, where fish, macro-, and megafauna are absent, hosts relatively large abundances of protists and micrometazoans[6].

Many pelagic zooplankton organisms have benthic stages and can survive hypoxic/anoxic conditions in the form of resting eggs[8,9]; such eggs have been shown to hatch once oxygen returns[10]. However, some eukaryotic organisms are adapted to live in anoxia, which may be due to the presence of copious organic matter and low predation pressure[6,11]. Nematodes are among the most abundant animals in these regions[12–14], and have evolved strategies to cope with low oxygen conditions[15,16]. However, adaptation and community responses of benthic organisms to oxygen starvation have only recently started to be investigated[6,17],

and the mechanism through which they survive long-term anoxia is one of the most intriguing questions in marine ecology.

Marine OMZs are oxygen limited, but only occasionally become euxinic (i.e., both absent in oxygen and rich in sulfide), except in rare cases when sulfate reduction becomes important under nitrate-limited conditions[18]. Enclosed marine basins (e.g., Baltic and Black Seas), receiving high loads of organic matter and with euxinic waters, host microbial communities largely thriving on sulfur metabolism[19]. These areas are considered inhospitable to aerobically respiring organisms, as the main product of sulfate reduction, i.e., hydrogen sulfide ($H_2S$), is toxic to aquatic life. Free $H_2S$ can lead to respiratory stress to benthic organisms already at micromolar concentrations[20], and at ca. 14 μmol $L^{-1}$, $H_2S$ effects on marine benthic organisms at a population level start arising[21]. However, certain aerobic organisms, including nematodes, gastrotrichs, and gnathostomulids, can live in sulfidic sediments[22]. Several nematode species can detoxify from sulfides by creating a viscous shield consisting of elemental sulfur in the epidermis[13,23]. Other nematode species live in symbiosis with sulfide-oxidizing bacteria, which may protect them from sulfide[24]. Under anoxic conditions and when nitrate is present, such bacteria are known to couple sulfide oxidation with nitrate reduction[25,26], and this process may yield oxidized nitrogen compounds such as nitrous oxide ($N_2O$)[25,26]. $N_2O$ has therefore been shown to be a good indicator of potential nitrate reduction at the oxic–anoxic interface of the Baltic Sea dead zone[27]. While microbial ecology studies in euxinic systems proliferate, there is a large knowledge gap concerning species diversity and potential metabolism of multicellular anaerobic eukaryotes. To our knowledge, there are no studies using RNA sequencing to analyze both rRNA and mRNA to investigate dead zone animals.

This study aimed to use molecular data to advance our understanding of micrometazoan diversity and metabolism in low oxygen and sulfidic environments. Specifically, we hypothesized that (1) low oxygen and high sulfide concentrations reduce metazoan diversity and alter community structure, and (2) mRNA transcripts translating for metazoan proteins in dead zone sediments (DZS) are significantly different (in amount and function) in response to oxygen, nitrous oxide, and sulfide concentrations.

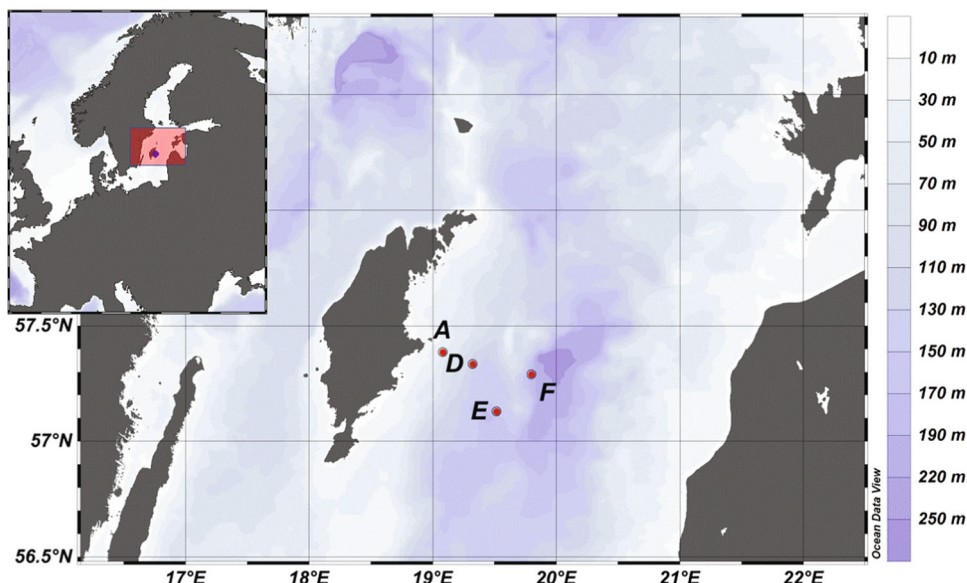

**Fig. 1 Location of the four sampling stations and bathymetry of the Baltic Proper.** Sediment cores and water samples were collected in April 2018 from each station indicated in the map. Sediments were either sectioned (0–2 cm sediment layer) for later molecular and microscopy analyses or kept intact and microprofiled onboard for porewater chemistry. Station A is 60-m deep and permanently oxygenated; Station D is 130-m deep and strictly hypoxic and sulfidic; Station E is 170-m deep, anoxic with $N_2O$; Station F is 210-m deep and anoxic.

To tackle these hypothesis, we conducted a sampling campaign in the central Baltic Sea (Fig. 1), the largest dead zone in the world[7]. We analyzed sediments with conditions of normoxia (>300 µmol $L^{-1}$ $O_2$), severe hypoxia (ca. 10 µmol $L^{-1}$ $O_2$), severe hypoxia/ anoxia (0–5 µmol $L^{-1}$ $O_2$), and complete anoxia (0 µmol $L^{-1}$ $O_2$). These DZS presented different availability of oxidized nitrogen (i.e., $N_2O$) and $H_2S$.

Here we show that DZS contain animal life adapted to cope with these harsh conditions. Alpha diversity and community structure based on rRNA data, differs significantly among anoxic and euxinic sites. Our results indicate that zooplanktons are present as resting stages in DZS, and the mRNA data suggest that these organisms use the enzyme cytochrome c oxidase (COX) as an oxygen sensor, which has previously been shown in, e.g., yeast[28]. In addition, nematodes can persist in anoxic and sulfidic sediments in niches like sulfide oxidation zones, or in low abundance potentially with a downregulated metabolism. To our knowledge, this is the first study using a comprehensive molecular dataset to study animals in dead zones. The findings imply that even on a low molecular level, dead zones might not be as dead as the terminology implies.

## Results

**Chemical environment characterization.** Water column profiles: The measured oxygen concentration in the water column was high (>400 µmol $L^{-1}$) vertically in the water column profile at Station A (Fig. 1). At the other three stations, the onset of a chemocline caused a sharp decrease in $O_2$ concentration between 65- and 70-m depth (Fig. 2). At 100-m depth, we recorded an oxygen pocket at stations D–F with concentrations 18-25 µmol $L^{-1}$ (Fig. 2). At stations D and E, traces of $O_2$ (<10 µmol $L^{-1}$) were detectable in the bottom water, whereas station F had bottom water anoxia (Fig. 2). $N_2O$ did not show any trend at A, while it clearly peaked at the depth of the oxygen pocket at the impacted stations. At station F, below the peak, $N_2O$ decreased monotonically with depth, whereas it showed a slight increase in concentrations at station E in proximity of the bottom.

Sediment microprofiles: Porewater microprofile measurements showed that $O_2$ was present at high concentrations (>300 µmol $L^{-1}$) at the sediment–water interface at station A (Fig. 2 and Table 1). Hypoxic conditions (8.8 µmol $L^{-1}$) and almost anoxic conditions (1.8 µmol $L^{-1}$) were recorded at the sediment–water interface at stations D and E, respectively. No $O_2$ was measured at station F. It cannot be excluded that minimal $O_2$ contamination happened during sampling and microprofiling at station E, although great care was taken to mimic in situ conditions. $O_2$ correlated negatively with $H_2S$ (rho = −0.78, $P < 0.001$) and positively with $N_2O$ (rho = 0.44, $P < 0.001$) in the measured sediment cores (tested for the whole dataset from all stations, Spearman correlations, Supplementary Data 1 and Supplementary Table 1).

Oxygen penetrated into the sediment to 7.0, 1.4, and 0.7 mm at stations A, D, and E, respectively (Fig. 2). High $N_2O$ concentrations (471 nmol $L^{-1}$) were recorded at the sediment–water interface at station E, where $N_2O$ penetrated to 3-mm depth (Fig. 2). Concentrations of $N_2O$ were two orders of magnitude lower at stations A (19 nmol $L^{-1}$) and F (29 nmol $L^{-1}$), and reached zero at 16- and 5-mm depth, respectively (Fig. 2). It was not possible to measure any $N_2O$ profile at station D. The highest porewater sulfide concentration was measured at station D (85 µmol $L^{-1}$ $H_2S$ at 1-cm depth). At this station, sulfide reached the sediment–water interface determining a zone where both $O_2$ and $H_2S$ were present (Fig. 2). At station E, $H_2S$ appeared below the oxic zone at 2-mm depth and reached 21 µmol $L^{-1}$ at 1-cm depth. At station F, $H_2S$ appeared at 0.8 mm, where 32 µmol $H_2S$ $L^{-1}$

was recorded at 1-cm depth. At station A, $H_2S$ was close to zero all the way down to 1-cm depth (Fig. 2).

**Metazoan diversity, community composition, and metabolism.** Eukaryotic diversity and community composition: The alpha diversity of the eukaryotic community composition in the 0–2 cm sediment layer, based on active taxa (i.e., 18S rRNA sequences), was different between stations ($n = 3$ per station, Fig. 3a). Full data are available in Supplementary Data 2 (SILVA taxonomy classifications), Supplementary Data 3 (NCBI NT taxonomy classifications), and Supplementary Table 2 (alpha diversity indexes). In more detail, station A had a higher alpha diversity (7.51 ± 0.06 Shannon's H) compared with the other stations (one-way ANOVA post hoc Tukey test, $P < 0.01$ for all tests, Fig. 3a). Furthermore, there was also a lower alpha diversity at stations D (5.03 ± 0.24 Shannon's H) and F (4.85 ± 0.23, $P < 0.01$) when compared with E (5.60 ± 0.04, $P < 0.05$, Fig. 3a). Nonmetric multidimensional scaling (NMDS) analysis of eukaryotic beta diversity showed that the stations formed different clusters, especially station A ($O_2$ rich and almost no $H_2S$), compared with the hypoxic–anoxic stations that all had higher concentrations of sulfide, when tested for the presence/absence and the relative abundance (PERMANOVA, Sørensen index, and Bray–Curtis dissimilarity, $F = 13.4$ and $F = 43.1$, respectively, $P < 0.01$ for both tests; Sørensen Fig. 3b, and Bray–Curtis in Supplementary Fig. 1). In the same analysis, station E that had the highest concentration of $N_2O$ clustered differently when compared with the other hypoxic–anoxic stations D and F. See Supplementary Fig. 2 for an overview of all eukaryotic phyla detected in the samples.

Looking closer at metazoan phyla, station A had a significantly higher relative abundance of Annelida (1.55 ± 0.91% in station A), Cnidaria (0.40 ± 0.03%), Kinorhyncha (0.49 ± 0.24%), Platyhelminthes (2.13 ± 0.61%), Priapulida (0.21 ± 0.03%), and Xenacoelomorpha (2.33 ± 0.47%) compared with the other stations (one-way ANOVA, post hoc Tukey test, all $P < 0.05$, Fig. 4a). In contrast, Arthropoda were significantly lower at station A compared with the other stations (12.09 ± 2.91% compared with stations D (38.47 ± 5.38%), E (30.13 ± 3.13%), and F (38.00 ± 2.97%), all $P < 0.01$, Fig. 4a). A similar pattern was observed for Rotifera, dominated by the class Monogononta, which had a higher relative abundance at stations D–F compared with A (Fig. 4a and Supplementary Data 2). The phylum Nematoda had the highest relative abundance at stations A and E. At station A, the relative abundance was 7.64 ± 0.55%, and was significantly higher compared with D (0.26 ± 0.05%) and F (0.46 ± 0.24%) ($P < 0.01$ for all tests, Fig. 4a). Similarly, station E also had a significantly higher relative abundance of Nematoda (5.43 ± 2.16%) compared with D and F (all $P < 0.01$, Fig. 4a). As Arthropoda, Rotifera, and Nematoda were the metazoan with the highest relative abundance in the sediment, data for these groups were analyzed further for community structure and metabolic functions.

Arthropoda and Rotifera taxonomy and metabolism: There was a significantly larger relative abundance of the cladoceran genus *Bosmina* (class Branchiopoda, phylum Arthropoda) at stations D (68.7 ± 1.1% 18S rRNA of Arthropoda), E (67.4 ± 1.4%), and F (66.0 ± 3.0%) compared with A (9.3 ± 1.4%) (one-way ANOVA post hoc Tukey test, $P < 0.01$, Fig. 4b). The cladoceran genus *Eubosmina* (former genus name of *Bosmina*) also had significantly higher relative abundance at stations D–F ($P < 0.01$, Fig. 4b). Rotifera was dominated by the class Monogononta, and genera *Synchaeta* (no significant difference between stations in relative abundance), and a higher relative abundance of *Keratella* at E compared with stations A and D ($P < 0.05$, Supplementary Data 2).

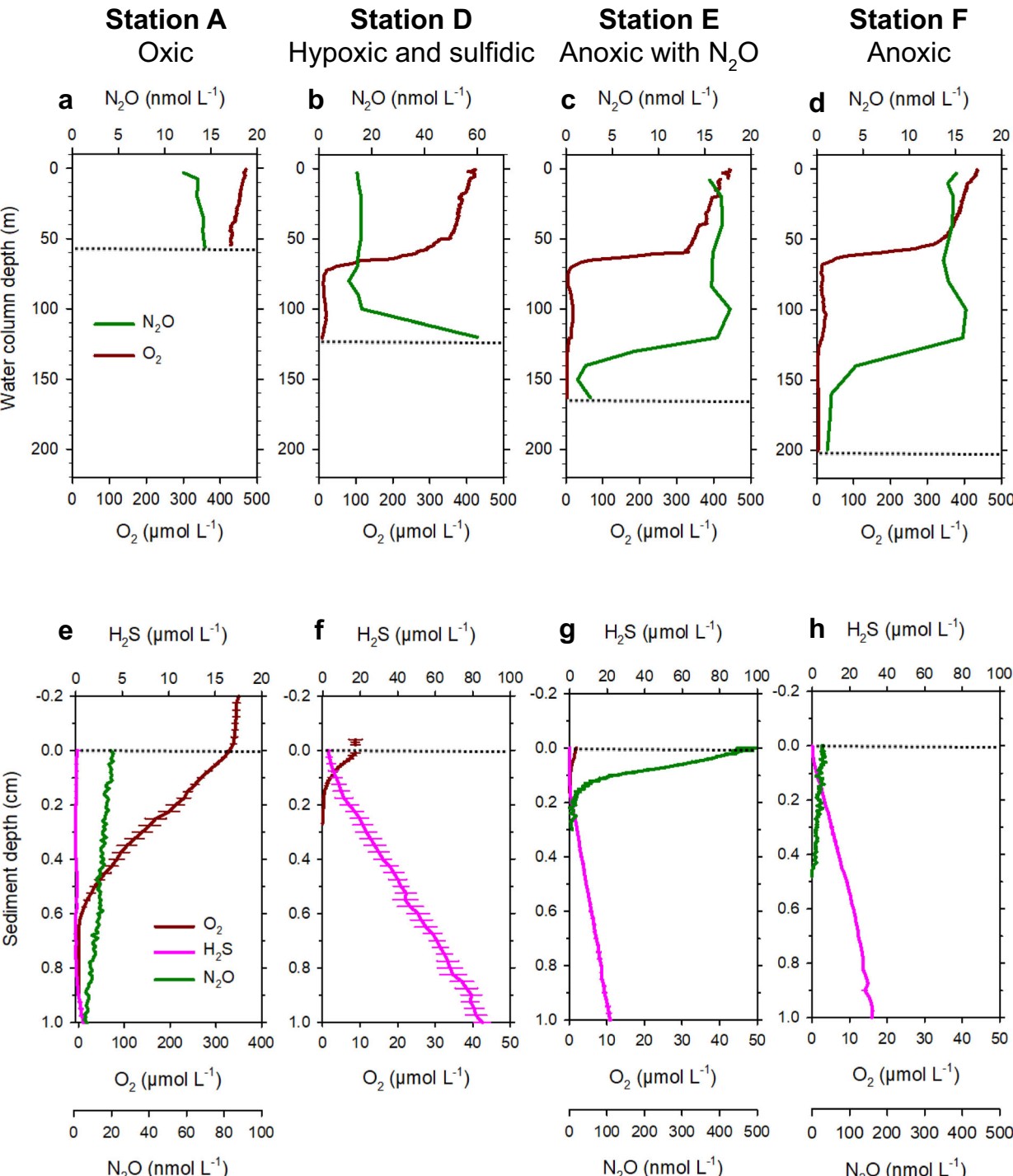

**Fig. 2 Water column and sediment profiles of $O_2$, $H_2S$, and $N_2O$.** Top four panels: vertical concentration profiles of oxygen ($O_2$) and nitrous oxide ($N_2O$) in the water column at station A (**a**), station D (**b**), station E (**c**), and station F (**d**). Bottom four panels: concentration microprofiles of oxygen ($O_2$), hydrogen sulfide ($H_2S$), and nitrous oxide ($N_2O$) in sediments of station A (**e**), station D (**f**), station E (**g**), and station F (**h**). Bold lines represent average microprofiles, and horizontal bars indicate standard error of the mean. The sediment–water interface is indicated by the horizontal dotted lines.

RNA transcripts successfully classified against the NCBI NR database and related to Arthropoda taxonomy showed significantly lower number of database hits for station A when compared with D (one-way ANOVA, post hoc Tukey test, $P <$ 0.05, Supplementary Data 4, Fig. 4d). Proteins affiliated with the family Bosminidae (including genera *Bosmina* and *Eubosmina*) at D–F were largely represented by aerobic respiration enzyme COX subunit I (IPR000883), and e.g., respiration chain enzyme NADH:ubiquinone oxidoreductase chain 2 (IPR003917) and

stress-related heat shock protein Hsp90 (IPR001404) (only four proteins affiliated for Bosminidae, Supplementary Table 3). Similarly, proteins affiliated with Rotifera at D–F were dominated by small heat shock protein HSP20 (IPR031107), COX subunit I (IPR000883), potassium channel inhibitor (IPR001947), and electron transport protein Cytochrome b (IPR030689) (17–134 proteins affiliated with Rotifera, Supplementary Table 4). These data indicate that Arthropoda and Rotifera animals were under stress in the hypoxic and anoxic sediments. The low molecular

**Table 1 Sediment microprofiling results for each station.**

| Parameter | Depth (cm) | A | D | E | F |
|---|---|---|---|---|---|
| $O_2$ (µM) | 0 | 329.7 ± 5.5 | 8.8 ± 1.3 | 1.9 ± 0.1 | 0 |
| | 0.5 | 35.0 ± 8.7 | 0 | 0 | 0 |
| | 1.5 | 0 | 0 | 0 | 0 |
| $H_2S$ (µM) | 0 | 0.2 ± 0.2 | 0.2 ± 0.3 | 0 | 0.0 ± 0.2 |
| | 0.5 | 0.2 ± 0.1 | 41.7 ± 8.0 | 8.3 ± 2.3 | 17.4 ± 0.7 |
| | 1.5 | 0.1 ± 0.1 | 106.4 ± 6.6 | 33.4 ± 3.7 | 40.7 ± 0.6 |
| $N_2O$ (nM) | 0 | 19.1 ± 3.4 | – | 471.0 ± 24.2 | 29.0 ± 1.0 |
| | 0.5 | 15.7 ± 1.8 | – | 0 | 0 |
| | 1.5 | 2.5 ± 1.4 | – | 0 | 0 |

The table shows $O_2$, $H_2S$, and $N_2O$ at three different depth layers starting at the sediment surface. The values show the mean ± SE ($n$ = 3–8 microprofiles per station). $N_2O$ data are missing for station D.

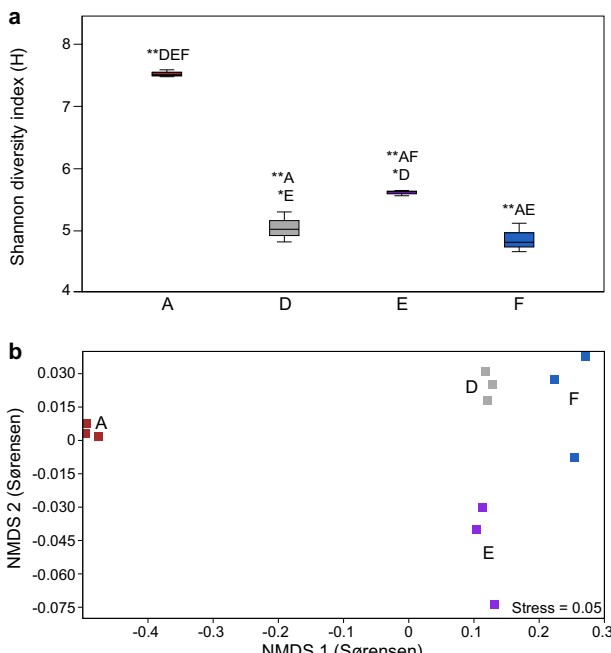

**Fig. 3 Eukaryotic alpha and beta diversity in the sediment at the different stations. a** Boxplot graphs showing the alpha diversity (Shannon's H) of the eukaryotic community in the top 2 cm sediment, based on the SILVA-classified RNA data (extracted 18S rRNA data, $n$ = 3 biologically independent samples per site). Statistically significant differences are denoted, * ($P$ < 0.05) and ** ($P$ < 0.01) followed by sampling sites that were different. The center line in the boxes represents the median; top and bottom lines of the boxes show the first and third quartiles. The top and bottom whiskers show the maximum and minimum values, respectively. **b** NMDS of the Sørensen index based on the presence/absence of the SILVA-classified 18S rRNA eukaryotic community composition for RNA samples. The colors denote sediment samples from stations A (brown), D (gray), E (purple), and F (blue).

activity in the RNA transcript dataset suggests that these animals were surviving in resting stages (such as dormancy or eggs).

Nematoda taxonomy and metabolism: The 18S rRNA data for nematodes showed a high diversity of genera over several families (Fig. 4c). Alpha diversity for nematodes was higher at stations A, D, and F (Shannon's H 4.1 ± 0.5) compared with station E (2.0 ± 0.5, one-way ANOVA post hoc Tukey test, $P$ < 0.05, Supplementary Table 2). At station F, the genus *Sabatieria* had a significantly higher relative abundance compared with A (33.4 ± 21.3% compared with 0.1 ± 0.1%, respectively, $P$ < 0.05,

Fig. 4c). The genus *Halomonhystera* had a higher relative abundance at E (73.9 ± 9.3% compared with <10.5% for the other stations, $P$ < 0.01 for all tests across stations, Fig. 4c). At station A, several genera belonging to different families had higher relative abundances compared with the other stations (e.g., families Axonolaimidae, Cyatholaimidae, Microlaimidae, and Xyalidae, $P$ < 0.01 when tested for genera *Axonolaimus*, *Cyatholaimus*, *Paracanthonchus*, *Calomicrolaimus*, and *Microlaimus*, respectively, Fig. 4c). Unclassified nematode 18S rRNA sequences had a high relative abundance at station D compared with the other stations ($P$ < 0.05, Fig. 4c).

RNA transcripts aligned against proteins in the NCBI NR database and linked to nematode taxonomy showed that station A had more database hits affiliated with nematodes (one-way ANOVA post hoc Tukey test, $P$ < 0.01), as well as station E compared with D and F ($P$ < 0.01, Fig. 4d). There were more proteins affiliated with Nematoda at A (310 ± 6 proteins, $P$ < 0.01), followed by E (170 ± 45 proteins, $P$ < 0.01). Stations D and F had a similar number of proteins (14 ± 6 and 21 ± 7, respectively) (Fig. 5). COX subunit I (IPR000883) had the highest counts per million sequence (CPM) values for all proteins at stations A and E, but was also present at D and F (Fig. 5). In stations D and F, the superfamily of proteolytic enzyme Peptidase C1A (IPR013128) had higher CPM values, as well as the Major facilitator superfamily (IPR002423), which includes proteins involved in membrane transport solutes (Fig. 5). Furthermore, the Chaperonin Cpn60/TCP-1 family (IPR002423) was higher at station D. Proteins involved in glycolysis included, e.g., pyruvate kinase and malate/L-lactate dehydrogenase, and these proteins were affiliated with nematodes in the hypoxic and anoxic sediments (stations D and E, Supplementary Data 4). Ribosomal proteins were available at all stations (Fig. 6). There was no detection of "transcription initiation" and "translation elongation factor" proteins at stations D and F, and the detection of RNA and DNA polymerases was also lower at the same stations (Fig. 6). In contrast, these essential proteins in gene transcription and protein translation were present at stations A and E (Fig. 6). Similarly, citrate synthase used in aerobic respiration was only detected at stations A and E (Supplementary Data 4).

**Microscopy visual identification of DZS metazoan**. In accordance with the molecular data, visual observation of samples confirmed the presence of a conspicuous number of Bosminidae-like resting stages in the anoxic sediment (Fig. 7a, see more photos in Supplementary Fig. 3). Microscopy analyses also confirmed the presence of nematodes *Halomonhystera* sp. (Fig. 7b), *Sabatieria* sp. (male Fig. 7c, female Fig. 7d, and juvenile Fig. 7e; Supplementary Fig. 4), and *Linhomoeidae* sp. (Fig. 7f).

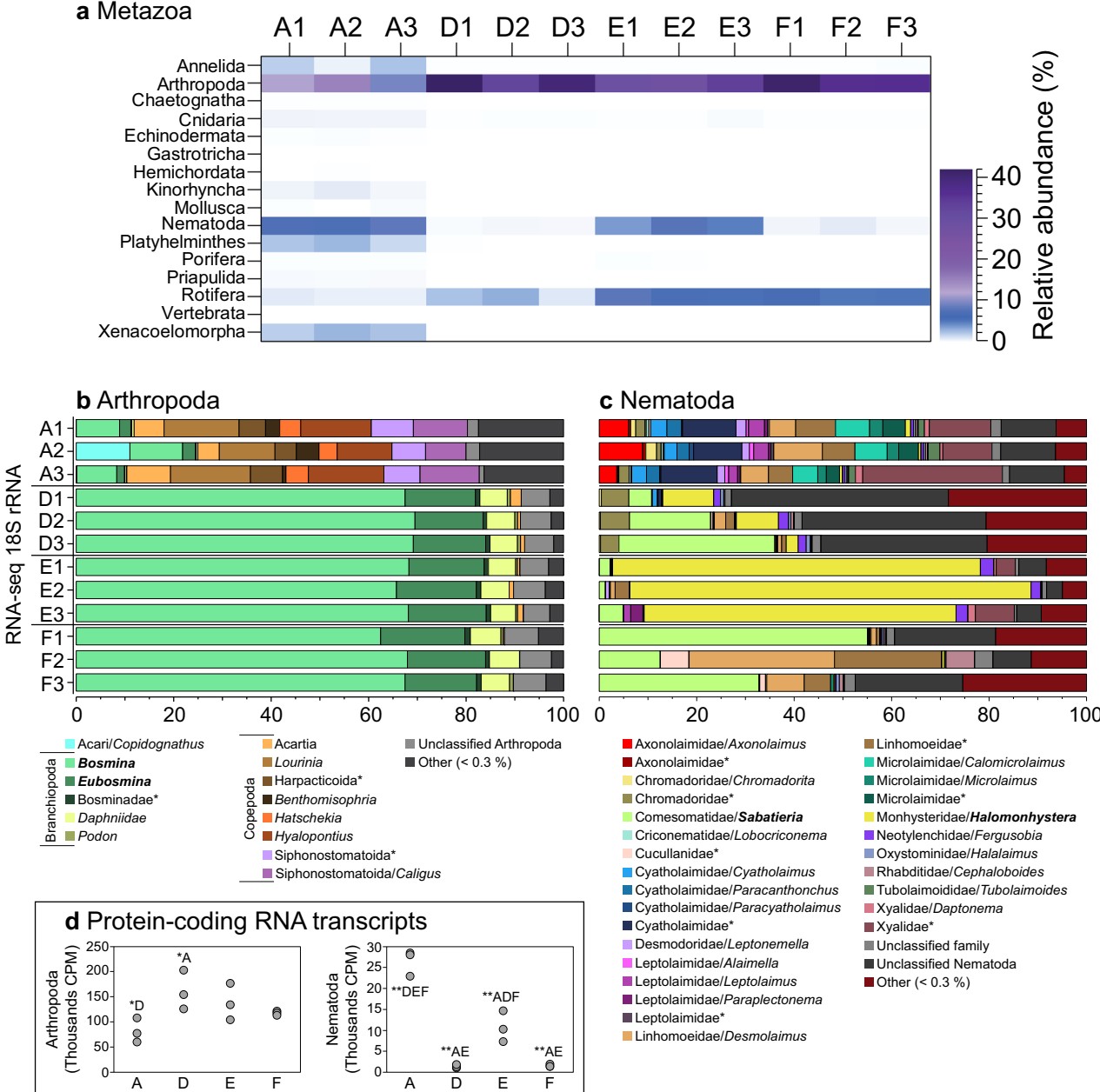

**Fig. 4 Metazoan community composition and activity in the sediment in each sample. a** Metazoan 18S rRNA community composition in the sediment based on extracted 18S rRNA sequences from the RNA-seq (SILVA database). The heatmap shows taxonomy groups >0.01% (average of all samples). The colors denote relative abundance with white representing 0%, white–blue gradient 0–6%, blue–purple gradient medium 6–12%, and light purple–dark purple gradient 12–42%. **b** Relative abundance of Arthropoda classes/genera, and **c** Nematoda families/genera based on the RNA-seq 18S rRNA sequences classified against the NCBI NT database. The *x* axis shows the relative abundance (%) for the Arthropoda and Nematoda phyla. Bold text denotes genera with a high relative abundance, while stars denote taxonomic classifications that could not be assigned to a genus for specific classes or families. **d** RNA transcripts for Arthropoda and Nematoda that were successfully classified against the NCBI NR database. The y axis shows the sum of normalized read counts as counts per million sequences (CPM) of all eukaryotic taxa. Significant statistical differences between sites are denoted. * ($P < 0.05$) and ** ($P < 0.01$) followed by sampling sites that were different.

## Discussion

This study provides the first attempt to uncover metabolic pathways and diversity of active animals in DZS using up-to-date sequencing techniques[29]. Dead zone conditions—i.e., $O_2$ concentration below 22 μmol $L^{-1}$—generally lead to mass mortality of animals[7]. The investigated deeper stations D–F had euxinic waters for several years before the inflow of salty, oxygenated North Sea water (major Baltic inflow), which increased bottom water $O_2$ levels to 10–50 μM between June 2015 and January

2017[30]. Since then, there were no more inflows. At the time of sampling, station F was anoxic (0 μmol $L^{-1}$ $O_2$), station E was anoxic to severely hypoxic (0–5 μmol $L^{-1}$ $O_2$), and station D was severely hypoxic (7–10 μmol $L^{-1}$ $O_2$). These sites have thus experienced dead zone conditions for at least 16 months continuously.

Nematodes had the highest diversity among metazoan taxa. In the sediment, organic material undergoes degradation and diagenesis[26]; thus, portions of the molecular data might derive from

| InterPro ID | Protein family/superfamily | A1 | A2 | A3 | D1 | D2 | D3 | E1 | E2 | E3 | F1 | F2 | F3 |
|---|---|---|---|---|---|---|---|---|---|---|---|---|---|
| IPR000883 | Cytochrome c oxidase subunit I | 47 | 37 | 47 | 0 | 0 | 91 | 159 | 103 | 113 | 45 | 44 | 0 |
| IPR013128 | Peptidase C1A | 16 | 6 | 2 | 182 | 130 | 45 | 27 | 5 | 13 | 23 | 59 | 158 |
| IPR011701 | Major facilitator superfamily | 9 | 8 | 6 | 212 | 0 | 0 | 7 | 7 | 3 | 45 | 44 | 0 |
| IPR002423 | Chaperonin Cpn60/TCP-1 family | 0 | 1 | 2 | 152 | 43 | 45 | 5 | 2 | 8 | 0 | 0 | 79 |
| IPR010067 | Aliphatic sulfonates-binding protein | 0 | 3 | 0 | 0 | 43 | 0 | 7 | 2 | 10 | 159 | 0 | 0 |
| IPR013126 | Heat shock protein 70 family | 12 | 15 | 3 | 91 | 0 | 0 | 22 | 11 | 5 | 0 | 59 | 0 |
| IPR001461 | Aspartic peptidase A1 family | 8 | 26 | 6 | 0 | 0 | 0 | 59 | 29 | 43 | 23 | 15 | 0 |
| IPR002347 | Short-chain dehydrogenase/reductase SDR | 18 | 28 | 11 | 0 | 0 | 0 | 10 | 18 | 18 | 91 | 0 | 0 |
| IPR002053 | Glycoside hydrolase, family 25 | 0 | 0 | 0 | 0 | 43 | 0 | 20 | 3 | 5 | 0 | 0 | 105 |
| IPR001580 | Calreticulin/calnexin | 3 | 0 | 6 | 0 | 43 | 0 | 0 | 0 | 0 | 0 | 103 | 0 |
| IPR000990 | Innexin | 20 | 7 | 3 | 0 | 43 | 0 | 0 | 3 | 3 | 45 | 29 | 0 |
| IPR000217 | Tubulin | 5 | 12 | 8 | 0 | 0 | 91 | 20 | 2 | 8 | 0 | 0 | 0 |
| IPR001664 | Intermediate filament protein | 12 | 10 | 6 | 0 | 0 | 0 | 24 | 31 | 48 | 0 | 0 | 0 |
| IPR037608 | Stromal interaction molecule | 0 | 0 | 0 | 0 | 130 | 0 | 0 | 0 | 0 | 0 | 0 | 0 |
| IPR001806 | Small GTPase superfamily | 7 | 10 | 14 | 30 | 0 | 45 | 0 | 15 | 5 | 0 | 0 | 0 |
| IPR000218 | Ribosomal protein L14P | 1 | 1 | 0 | 30 | 0 | 0 | 2 | 5 | 0 | 23 | 0 | 53 |
| IPR001534 | Transthyretin-like | 19 | 19 | 9 | 0 | 0 | 0 | 17 | 25 | 18 | 0 | 0 | 0 |
| IPR001392 | Clathrin adaptor, mu subunit | 3 | 3 | 3 | 0 | 0 | 23 | 0 | 0 | 3 | 68 | 0 | 0 |
| IPR026610 | 3'-RNA ribose 2'-O-methyltransferase, Hen1 | 0 | 0 | 3 | 0 | 87 | 0 | 0 | 11 | 0 | 0 | 0 | 0 |
| IPR002610 | Peptidase S54, rhomboid | 0 | 1 | 2 | 0 | 0 | 0 | 0 | 0 | 5 | 91 | 0 | 0 |
| IPR001697 | Pyruvate kinase | 4 | 1 | 5 | 0 | 87 | 0 | 0 | 2 | 0 | 0 | 0 | 0 |
| IPR002293 | Amino acid/polyamine transporter I | 1 | 1 | 0 | 0 | 0 | 91 | 0 | 0 | 0 | 0 | 0 | 0 |
| IPR010232 | Probable protein kinase UbiB | 0 | 0 | 2 | 91 | 0 | 0 | 0 | 0 | 0 | 0 | 0 | 0 |
| IPR005292 | Multi drug resistance-associated protein | 1 | 0 | 3 | 0 | 87 | 0 | 0 | 0 | 0 | 0 | 0 | 0 |
| IPR002213 | UDP-glucuronosyl/UDP-glucosyltransferase | 5 | 0 | 2 | 0 | 0 | 0 | 0 | 0 | 5 | 0 | 74 | 0 |
| IPR002587 | Myo-inositol-1-phosphate synthase | 4 | 0 | 0 | 30 | 0 | 0 | 2 | 0 | 3 | 45 | 0 | 0 |
| IPR000276 | G protein-coupled receptor, rhodopsin-like | 12 | 8 | 8 | 0 | 0 | 23 | 17 | 7 | 10 | 0 | 0 | 0 |
| IPR024704 | Structural maintenance of chromosomes protein | 0 | 0 | 0 | 0 | 0 | 0 | 2 | 0 | 0 | 0 | 0 | 79 |
| IPR010139 | Imidazole glycerol phosphate synthase, subunit H | 0 | 0 | 2 | 0 | 0 | 0 | 0 | 0 | 0 | 0 | 0 | 79 |
| IPR017071 | Transcription elongation factor Spt5 | 0 | 0 | 0 | 0 | 0 | 0 | 0 | 0 | 0 | 0 | 0 | 79 |
| IPR001486 | Truncated hemoglobin | 0 | 0 | 0 | 0 | 0 | 0 | 0 | 0 | 0 | 0 | 0 | 79 |
| IPR001019 | Guanine nucleotide binding protein (G-protein), alpha subunit | 4 | 7 | 6 | 0 | 0 | 0 | 0 | 0 | 0 | 0 | 59 | 0 |
| IPR015720 | TMP21-related | 1 | 1 | 2 | 0 | 43 | 0 | 0 | 0 | 0 | 0 | 0 | 26 |
| IPR016685 | RNA-induced silencing complex, nuclease component Tudor-SN | 0 | 0 | 0 | 0 | 0 | 0 | 0 | 0 | 0 | 0 | 74 | 0 |
| IPR001723 | Nuclear hormone receptor | 3 | 1 | 0 | 0 | 0 | 68 | 0 | 0 | 0 | 0 | 0 | 0 |
| IPR025778 | Histone-lysine N-methyltransferase, EZ | 0 | 0 | 0 | 0 | 0 | 68 | 0 | 0 | 0 | 0 | 0 | 0 |
| IPR015655 | Protein phosphatase 2C family | 4 | 7 | 0 | 0 | 0 | 45 | 5 | 7 | 0 | 0 | 0 | 0 |
| IPR001404 | Heat shock protein Hsp90 family | 4 | 3 | 2 | 0 | 0 | 0 | 5 | 0 | 8 | 45 | 0 | 0 |
| IPR008209 | Phosphoenolpyruvate carboxykinase, GTP-utilising | 7 | 3 | 6 | 0 | 0 | 0 | 5 | 13 | 30 | 0 | 0 | 0 |
| IPR006515 | Polyadenylate binding protein, human types 1, 2, 3, 4 | 0 | 0 | 0 | 0 | 0 | 0 | 0 | 0 | 10 | 0 | 0 | 53 |
| | Total number of proteins classified | 318 | 307 | 312 | 12 | 16 | 24 | 143 | 232 | 152 | 22 | 30 | 17 |

CPM: 0 / 100 / 200

**Fig. 5 Nematoda RNA transcripts in the sediment identified with the InterPro database.** The heatmap was delimited to the top 40 proteins (average of all samples). The blue color gradient shows thousands of CPM for the phyla Nematoda (i.e., CPM × $10^{-3}$). The last row shows the number of classified proteins.

| InterPro ID | Protein family | A1 | A2 | A3 | D1 | D2 | D3 | E1 | E2 | E3 | F1 | F2 | F3 |
|---|---|---|---|---|---|---|---|---|---|---|---|---|---|
| IPR020609 | Archaeal RpoH /eukaryotic RPB5 RNA polymerase subunit | 0 | 0 | 0 | 0 | 0 | 0 | 0 | 0 | 0 | 0 | 29412 | 0 |
| IPR002298 | DNA polymerase A | 1350 | 0 | 0 | 0 | 0 | 0 | 0 | 0 | 0 | 0 | 0 | 0 |
| IPR024826 | DNA polymerase delta/II small subunit family | 1350 | 0 | 1580 | 0 | 0 | 0 | 0 | 0 | 0 | 0 | 0 | 0 |
| IPR001001 | DNA polymerase III, beta sliding clamp | 0 | 0 | 0 | 0 | 0 | 0 | 0 | 1634 | 0 | 0 | 0 | 0 |
| IPR012763 | DNA polymerase III, subunit gamma/ tau | 0 | 0 | 1580 | 0 | 0 | 0 | 0 | 0 | 0 | 0 | 0 | 0 |
| IPR022880 | DNA polymerase IV | 0 | 0 | 1580 | 0 | 0 | 0 | 0 | 0 | 0 | 0 | 0 | 0 |
| IPR006172 | DNA-directed DNA polymerase, family B | 0 | 0 | 0 | 0 | 0 | 0 | 0 | 0 | 0 | 0 | 0 | 26316 |
| IPR012164 | DNA-directed RNA polymerase subunit/transcription factor S | 2699 | 0 | 0 | 0 | 0 | 0 | 0 | 1634 | 10050 | 0 | 0 | 0 |
| IPR015712 | DNA-directed RNA polymerase, subunit 2 | 0 | 0 | 0 | 0 | 0 | 0 | 2439 | 0 | 0 | 0 | 0 | 0 |
| IPR009668 | RNA polymerase I associated factor, A49-like | 0 | 0 | 1580 | 0 | 0 | 0 | 0 | 0 | 0 | 0 | 0 | 0 |
| IPR037685 | RNA polymerase RBP11 | 1350 | 1381 | 0 | 0 | 0 | 0 | 4878 | 0 | 0 | 0 | 0 | 0 |
| IPR005574 | RNA polymerase subunit RPB4/RPC9 | 0 | 0 | 3160 | 0 | 0 | 0 | 0 | 0 | 0 | 0 | 0 | 0 |
| IPR008851 | Transcription initiation factor IIF, alpha subunit | 0 | 0 | 0 | 0 | 0 | 0 | 2439 | 0 | 0 | 0 | 0 | 0 |
| IPR003196 | Transcription initiation factor IIF, beta subunit | 5398 | 0 | 0 | 0 | 0 | 0 | 0 | 0 | 0 | 0 | 0 | 0 |
| IPR003162 | Transcription initiation factor TAFII31 | 0 | 1381 | 0 | 0 | 0 | 0 | 0 | 0 | 0 | 0 | 0 | 0 |
| IPR037794 | Transcription initiation factor TFIID subunit 12 | 0 | 0 | 1580 | 0 | 0 | 0 | 0 | 0 | 0 | 0 | 0 | 0 |
| IPR037815 | Transcription initiation factor TFIID subunit 3 | 0 | 0 | 1580 | 0 | 0 | 0 | 0 | 0 | 0 | 0 | 0 | 0 |
| IPR001253 | Translation initiation factor 1A (eIF-1A) | 0 | 1381 | 3160 | 0 | 0 | 0 | 0 | 0 | 0 | 0 | 0 | 0 |
| IPR001288 | Translation initiation factor 3 | 5398 | 6906 | 0 | 0 | 0 | 0 | 0 | 3268 | 2513 | 0 | 0 | 0 |
| IPR001040 | Translation Initiation factor eIF- 4e | 1350 | 2762 | 0 | 0 | 0 | 0 | 0 | 8170 | 2513 | 0 | 0 | 0 |
| IPR027512 | Eukaryotic translation initiation factor 3 subunit A | 2699 | 0 | 1580 | 0 | 0 | 0 | 0 | 0 | 0 | 0 | 0 | 0 |
| IPR027516 | Eukaryotic translation initiation factor 3 subunit C | 0 | 1381 | 0 | 0 | 0 | 0 | 0 | 0 | 0 | 0 | 0 | 0 |
| IPR007783 | Eukaryotic translation initiation factor 3 subunit D | 1350 | 0 | 0 | 0 | 0 | 0 | 0 | 0 | 2513 | 0 | 0 | 0 |
| IPR016650 | Eukaryotic translation initiation factor 3 subunit E | 0 | 0 | 1580 | 0 | 0 | 0 | 0 | 0 | 2513 | 0 | 0 | 0 |
| IPR027531 | Eukaryotic translation initiation factor 3 subunit F | 0 | 1381 | 0 | 0 | 0 | 0 | 0 | 0 | 0 | 0 | 0 | 0 |
| IPR027524 | Eukaryotic translation initiation factor 3 subunit H | 0 | 0 | 0 | 0 | 0 | 0 | 0 | 3268 | 0 | 0 | 0 | 0 |
| IPR027528 | Eukaryotic translation initiation factor 3 subunit M | 0 | 0 | 0 | 0 | 0 | 0 | 0 | 1634 | 0 | 0 | 0 | 0 |
| | Ribosomal proteins (sum of 46 proteins, CPM) | 33738 | 33149 | 36335 | 30303 | 86957 | 22727 | 56098 | 57190 | 45226 | 68182 | 58824 | 52632 |

CPM: 0 / 5000 / 10000

**Fig. 6 Nematoda RNA transcripts that were attributed to polymerases, transcription initiation factors, translation initiation factors, and ribosomal proteins.** The green color gradient in the heatmap shows CPM for the phylum Nematoda. The last row shows the CPM values for ribosomal proteins.

damaged ribosomes or degraded hereditary material. However, the presence of RNA transcripts (i.e., mRNA) strongly indicates that some nematode species were alive and metabolically active in this DZS. Nematodes are known to tolerate hypoxia[15,31,32], and have been observed, e.g., in the Gulf of Mexico and Black Sea dead zones[33,34,35]. Benthic nematodes can temporarily cope with anoxia by migrating upward to the overlying oxic water until normoxic conditions return to the sediment[32]. However, at the sites here studied, 60–140-m migration would be extremely difficult to achieve, and would not explain why the nematodes were detected in the sediment. It is more likely that benthic nematodes were adapted and able to survive in the oxygen-deficient

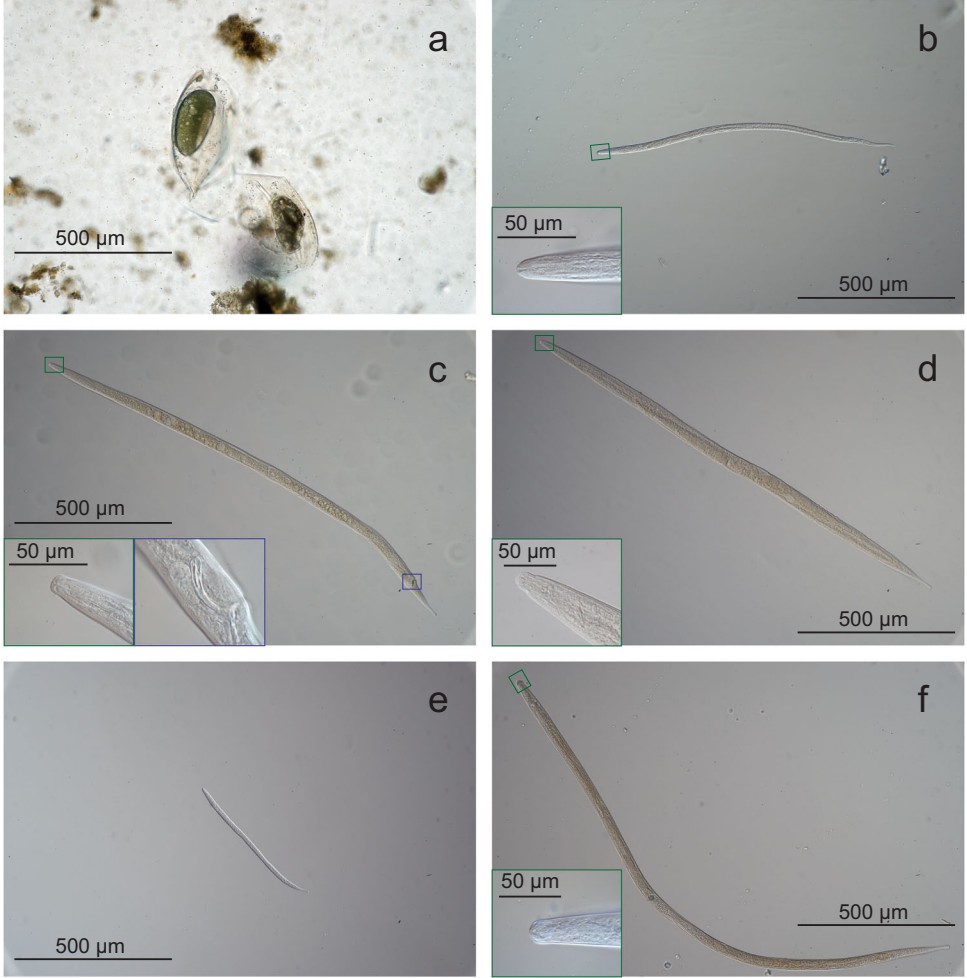

**Fig. 7 Microscopy images of Bosminidae-like organisms and nematodes from station E. a** Bosminidae-like resting eggs. **b** Juvenile *Halomonhystera* with the inset showing a higher magnification of the buccal cavity (green frame). **c** Male *Sabatieria* sp. with the inset of buccal cavity (green) and copulatory spicules (blue). **d** Female *Sabatieria* sp. with the inset of buccal cavity (green). **e** Juvenile *Sabatieria* sp. **f** Juvenile *Linhomoeidae* with the inset of buccal cavity (green). Scale bars are 500 μm and 50 μm for insets.

conditions. It has been proposed that the quantity of food for benthic fauna is usually high in oxygen-deficient zones, which together with complete absence of larger predators, would make these organic-rich sediments suitable for colonization of certain micrometazoans[6]. Nematodes are among the micrometazoan groups that have successfully evolved to cope with anoxia and sulfides[23,24,36]. For example, short time exposure to hypoxia (up to 7 days) had negligible effects on various nematode species[37], while 14 days of anoxia decreased the general abundance, but species such as *Sabatieria pulchra* showed resistance[38]. Furthermore, Taheri et al.[39] observed nematodes persisting in anoxic sediment after 307 days, including species belonging to the genus *Sabatieria*. Our results also showed that this genus was among the dominant nematodes at the hypoxic–anoxic stations in the 18S rRNA dataset, and found visually with microscopy.

Under such extreme conditions, certain nematodes are able to change to anaerobic metabolism (fermentation) or into a low metabolic state called cryptobiosis (reviewed in Tahseen[40]). Considering the lower number of sequences and the absence of essential enzymes for transcription and translation at stations D and F, it is possible that nematode communities at these stations consisted of low abundant taxa adapted or trying to survive in these extreme conditions. The strong difference in classified proteins further indicates that the metabolic activity was different at stations A and E compared with D and F. Furthermore, proteins affiliated with nematodes in the oxygen-deficient sediments, such as pyruvate kinase and malate/L-lactate dehydrogenase, were likely involved in anaerobic metabolisms[41]. Interestingly, citrate synthase was detected not only at the oxic station A, but also at the oxygen-deficient station E, which suggests that nematodes were able to use oxygen at extremely low concentrations. Previous studies have shown that as little as 17.6 μmol L$^{-1}$ O$_2$ can support aerobic respiration in nematodes from natural springs[42]. Our study indicates that nematodes might be able to respire aerobically at even lower oxygen concentrations ($\leq$1.8 μmol L$^{-1}$). Even though oxygen was present at station D, the nematode metabolic activity was lower than that at station A or E, which suggests that the high sulfide concentrations at station D might have had a detrimental effect on the nematode populations. However, nematode taxa belonging to the genus *Sabatieria* were found in the presence of high sulfide too, suggesting that these animals must have evolved efficient sulfide detoxification mechanisms[40].

A striking pattern in our results was the high relative abundance of the genus *Halomonhystera* at station E also confirmed visually with microscopy. This genus has previously been reported from bacterial mats at 1280-m water depth in sulfide-rich sediments[43]. It is therefore likely that bacterial denitrification

coupled to sulfide oxidation[26] in the sediment at station E (as indicated by the clear overlap between the N₂O and H₂S profiles at 2–3-mm depth) formed a niche habitat for *Halomonhystera*. The number of Nematoda taxa able to occupy such niche is small, which may also explain the lower diversity of nematodes at station E. Sulfide-oxidizing bacteria have previously been detected on nematode's (Stilbonematinae) cuticle, and this ectosymbiosis likely helps to detoxify high levels of sulfide[36,44]. Other nematodes (Oncholaimidae) are non-symbiotic and show detoxification through secreting epidermis inclusions made of elemental sulfur[23]. Interestingly, nematodes also had the lowest diversity at station E, and this was possible due to a specific niche of survival in this sediment dominated by a few species such as *Halomonhystera* and *Sabatieria*.

In the hypoxic/anoxic sediments, a predominant portion of RNA transcripts, affiliated with pelagic taxa like *Bosmina* (formerly *Eubosmina*) and Rotifera, was attributed to COX subunit I. This protein can be used as an oxygen sensor as seen for mammalian tissue cells[45] and yeast[28]. In addition, under anoxic conditions, COX functions as a nitrite reductase that produces nitric oxide in eukaryotic mitochondria[46]. The high number of 18S rRNA sequences and microscope observation of resting stages, but the limited number of RNA transcript-classified proteins detected for *Bosmina*, indicate that these populations consisted of resting eggs. Diapausing eggs of *Bosmina* have been found to be viable for 15–21 years[47], and possibly an egg bank in the sediment has accumulated over several years in the central Baltic Sea. Rotifer populations in the hypoxic/anoxic sediments had, in addition to COX subunit I, a major portion of the RNA transcripts attributed to the small heat shock protein HSP20. These shock proteins are upregulated in rotifer resting eggs[48], and such eggs have been observed to be viable for up to 100 years[47]. Zooplankton egg banks (including rotifer) have previously been observed in Baltic Sea anoxic sediments, and these eggs hatched upon oxygenation[10]. Rotifers can tolerate low oxygen conditions[49,50], and change to anaerobic metabolism during a few days up to a month[51,52]. Considering that there was a higher relative abundance of 18S rRNA sequences of rotifers at stations D–F compared with the oxic sediment at station A, it is likely that rotifer resting eggs were abundant and kept accumulating in the oxygen-deficient sediments, free from benthic predation, for a relatively long period of time. The enzyme citrate synthase—a proxy for aerobic metabolism[53]—was not present in either the *Bosmina* or Rotifera datasets at any station, further suggesting that these populations were dormant. Our results thus indicate that there is an available egg bank of zooplankton in sediments of the largest dead zones in the world. To our knowledge, this is the first study to indicate that dormant zooplankton uses COX subunit I as an oxygen sensor to cue for hatching.

To summarize, we have here shown that the diversity and community structure of metazoans in DZS are different between low oxygen areas, and that this is likely related to the concentration of sulfide in the sediment. Nematodes survive in specialized niches such as sulfide oxidation zones, or are in low abundance (potentially with a downregulated metabolism) in anoxic and sulfidic sediments. This was also indicated by the number of proteins classified to nematodes that were the highest in oxic and hypoxic sediments (sulfide oxidizing), when compared with sulfidic hypoxic and anoxic sediments. It has previously been shown that zooplankton eggs accumulate in anoxic sediment, and oxygen is a cue for hatching[10]. Our data further indicate that COX subunit I might be the key protein for sensing oxygen by the zooplanktonic dormant community. Reoxygenation of dead zones would therefore increase the flux of carbon to the water column, and thus enhance the benthic–pelagic coupling[10]. Moreover, nematode communities come back quickly

after the onset of normoxia[32,39,54], and would therefore increase the availability of food for recolonization of benthic communities. We conclude that animals are alive and adapted to survive in dead zones, and these biological resources are therefore not lost and could be important in the recovery of benthic metazoan communities if oxygen conditions improve.

## Methods

**Study area and sampling.** The central Baltic Sea is characterized by permanent thermohaline stratification in its deeper basins[27,55]. Two large inflows of saline and oxygenated waters reached the inner Baltic Sea in 2003 and 2014, triggering oxidation processes in otherwise anoxic bottom waters. Oxygenation is however an ephemeral event in the Baltic as the inflow of denser water masses leads to even stronger stratification[56]. For this study, we visited four stations (A, D, E, F) along a gradient of depth and bottom water oxygen concentrations in April 2018 onboard of R/V Skagerak (Fig. 1). Station A is 60-m deep and permanently oxygenated (sampled April 25, long 19°04′951, lat 57°23′106); Station D is 130-m deep and strictly hypoxic (O₂ ≤ 25 μmol L⁻¹, sampled April 26, long 19°19′414, lat 57°19′671); Station E is 170-m deep, anoxic, and nitrate-containing (sampled April 23, long 19°30′451, lat 57°07′518); Station F is 210-m deep, euxinic, and nitrate-free (sampled April 23, long 19°48′035, lat 57°17′225).

Water column oxygen profiles were measured by means of a CTD-rosette system (SBE 911plus, SeaBird Electronics, USA) equipped with O₂ sensors (SBE 43 Dissolved O₂ Sensor, SeaBird Electronics, USA). Water column sampling was carried out at different depths (n = 12) depending on the site water column height. The bottles from the CTD rosette were sampled immediately after withdrawal by means of a Viton© tubing, and subsamples for nitrous oxide (N₂O) were collected in 12-mL Exetainers (Labco, UK). The water was allowed to overflow for at least three times the Exetainer volume, biological activity stopped with 100 μL of a 7 mol L⁻¹ ZnCl₂ solution, and Exetainers immediately closed air-tight, stored upside down and refrigerated until later analysis. Analysis of N₂O was performed by the headspace technique on a gas chromatograph (SRI 8610C) equipped with an electron capture detector (ECD) using N₂ as carrier gas.

Sediment was collected with a modified box corer, which allows sampling of undisturbed surfaces even in very soft and highly porous sediments[57]. Two to three box core casts were done at each station, and up to nine PVC cylinders (5-cm diameter, 30-cm length) were subsampled in total. Three of these sediment cores were immediately processed for later nucleic acid extraction, while the rest of the sediment cores were transferred into an aquarium for sediment microprofiling (see below). Each sediment core used to extract RNA was quickly moved onto a sterile bench. The sediment was gently extruded, and the top 0–2-cm slice was directly transferred into a sterile 50-mL centrifuge tube, which was snap frozen in liquid N₂. Sediment slice samples (n = 12) were transferred from the seafloor to the liquid N₂ container within 15–20 min.

**Sediment microprofiling.** The bottom water in the aquarium was kept at in situ oxygen and temperature (ranging between 3.8 and 7.4 °C depending on station), by circulating water with a cooling unit (Julabo, DE), and by flushing it with a mixture of air and N₂/CO₂. Sediment microprofiles for dissolved oxygen (O₂), hydrogen sulfide (H₂S), and nitrous oxide (N₂O) concentrations were measured following the protocol illustrated by Marzocchi et al.[30]. Clark-type gas microsensors for O₂, H₂S, and N₂O were specifically built at Aarhus University (Denmark)[58–60]. At each station, three to five microprofiles were measured in each replicate core (n = 2–3 for stations A, D, and F, and n = 1 for station E) by mounting the microsensors onto a motorized micromanipulator (MM33, Unisense, Denmark), and recording vertical profiles with a four-channel multimeter (Unisense, Denmark) communicating with a laptop. Profiles for O₂ and N₂O were measured at a vertical resolution of 50–100 μm, while H₂S profiles were made using a vertical resolution of 250 μm. A water column of ~5 cm above the sediment was circulated by a gentle flow of air (station A) or N₂ (stations D–F) toward the water surface with a 45° angle. This allowed to maintain a constant diffusive boundary layer during measurements. Before each core was measured, the O₂ sensor was calibrated using a two-point calibration procedure in O₂-saturated bottom water (100% O₂) and ca. 1 cm inside the sediment (0% O₂). The H₂S sensor was calibrated in fresh anoxic solutions containing increasing amounts of a 10 mM Na₂S stock solution. The N₂O sensor was calibrated in N₂O-free water and in N₂O-amended water prepared by adding defined volumes of N₂O-saturated water to defined volumes of N₂O-free water.

**Nucleic acid extraction and sequencing.** RNA was extracted from ~2 g of thawed sediment following the RNeasy PowerSoil kit (QIAGEN). Sediment was thawed and homogenized but still cold when added into the bead and lysis solution. Extracted RNA was DNase treated with the TURBO DNA-free kit (Invitrogen), and was followed by ribosomal RNA depletion using the bacterial version of the RiboMinus Transcriptome Isolation Kit (ThermoFisher Scientific). Quantity and quality of extracted nucleic acids were measured on a NanoDrop One spectrophotometer (ThermoFisher Scientific). The RNA samples were confirmed to be free of DNA contamination using a 2100 Bioanalyzer (Agilent). Library

preparation of RNA for sequencing was prepared with the TruSeq RNA Library Prep v2 kit skipping the poly-A selection step (Illumina). The RNA was sequenced on one Illumina NovaSeq6000 S4 lane with a paired-end 2 × 150-bp setup at the Science for Life Laboratory, Stockholm.

**Microscopy visual identification**. The remaining thawed sediments from station E ($n = 3$) were diluted in an isotonic solution of NaCl in distilled water, and manually sorted under the Nikon SMZ1000 microscope with ×8 to ×80 magnification. All detected nematodes were fixed in isotonic 4% formaldehyde solution for a minimum of 2 days, processed to absolute glycerin following standard protocols[61], and mounted on permanent slides. Light microscopy photographs were taken using a Sony A7 mirrorless camera mounted on a Nikon Eclipse 80i microscope with differential interference contrast.

**Sequencing output and quality trimming**. RNA sequencing yielded on average 81.7 million read pairs per sediment sample ($n = 12$ with $n = 3$ per site). Illumina adapters were removed from the raw.fastq sequences by using SeqPrep 1.2[62], PhiX sequences were removed by mapping the reads against the PhiX genome (NCBI Reference Sequence: NC_001422.1) using bowtie2 2.3.4.3[63]. Quality trimming of the reads was conducted with Trimmomatic 0.36[64] with the following parameters: LEADING:20 TRAILING:20 MINLEN:50. Final quality of the trimmed reads was checked with FastQC 0.11.5[65] in combination with MultiQC 1.7[66]. After quality trimming, an average of 81.2 (min 73.0, max 87.9) million read pairs remained, with an average length of 144 bp. A full list with e.g. sequence facility labels, number of sequences before and after quality trimming, and number of extracted rRNA sequences is available in Supplementary Data 5.

**Taxonomic annotation**. Taxonomic annotation of the quality-trimmed reads was performed by first extracting SSU rRNA sequences using SortMeRNA 2.1b with the supplied SILVA reference database[67], followed by annotation using Kraken2 2.0.7[68]. Kraken2 was run using default settings with a paired-end setup against the small-subunit SILVA v132 NR99[69] and NCBI NT databases (databases downloaded: 1 and 12 March 2019, respectively). Both SILVA and NCBI NT were used due to database limitations for nematodes using SILVA (see, e.g., Holovachov et al.[70,71]). The Kraken2 output reports were combined into a biom-format file using the python package kraken-biom 1.0.1 (with the following setup:—fmt hdf5 -max D–min S). The biom-format file was then converted to a text table using the python package biom-format 2.1.7[72] and used for further downstream analyses. To remove uncertainty in the dataset taxonomic classifications, less than ten sequence counts were removed. The final 18S rRNA data yielded on average 5,369,739 sequences (SILVA classifications) and 4,177,670 sequences (NCBI NT classifications). The final taxonomy results were normalized between stations as relative abundance (%), and analyzed further in the software Explicet 2.10.5[73]. When visualizing lower taxonomic levels (Fig. 4b–c), freshwater and terrestrial taxa that were likely derived from database errors (Artropoda: *Teloganopsis*, *Stenchaetothrips*, and *Metatrichoniscoides*, and Nematoda: *Fictor* and *Strongyloides*) were included in the group "unclassified". A full list of all classifications is available in Supplementary Data 3.

**Protein classification of RNA transcripts**. Here we followed a bioinformatics protocol closely resembling the SAMSA2 pipeline[74] that uses the DIAMOND + MEGAN approach to classify non-rRNA-merged paired-end reads against a protein database[75]. Paired-end RNA sequences were merged using PEAR 0.9.10[76] (~75% merging rate), and SortMeRNA was used to extract non-rRNA-merged reads. This was followed by protein annotation against NCBI NR (database downloaded 2 April 2019) using the aligner software Diamond 0.9.10[77] in conjunction with BLASTX with an e-value threshold of 0.001. The diamond output files were analyzed in MEGAN 6.15.2[78] for taxonomy using default LCA parameters (NCBI taxonomy database: prot_acc2tax-Nov2018X1.abin) and protein annotation (NCBI NR accession linked to the InterPro database: acc2interpro-June2018X.abin) with databases available with MEGAN. The results of animals, indicated to be alive and active from the 18S rRNA sequence data, were then extracted from MEGAN and analyzed further. Sequence counts were normalized among samples as counts per million sequences (CPM, relative proportion ×1,000,000).

**Statistics**. Shannon's H alpha diversity index for the taxonomy data was calculated in Explicet after subsampling read counts to the lowest sample size (Eukaryota SILVA: 4,312,510 counts, Eukaryota NCBI NT: 3,486,047 counts, and Nematodes NCBI NT: 12,776 counts). NMDS multivariate analysis and PERMANOVA tests (9999 permutations) were conducted in the software Past 3.22[79]. Statistics of taxonomic data was conducted using SPSS 25, and Shapiro–Wilk tests were used to test for normal distribution. Differences between alpha diversity and phylogenetic groups were then tested using one-way ANOVA and post-hoc Tukey tests. All statistical tests are available in Supplementary Data 6.

**Reporting summary**. Further information on research design is available in the Nature Research Reporting Summary linked to this article.

## Data availability

The data that support these findings are available in the paper and supplementary files. The raw sequence data have been deposited online and can be accessed at the NCBI BioProject PRJNA531756.

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

## Acknowledgements

Individual financial support was provided by the Swedish Research Council Formas to SB (Grant no. 2017-01513); the Stockholm University's strategic funds for Baltic Sea research to F.N.; the Swedish Research Council Formas to F.N. (Grant no. 2016-00804);

the Swedish Environmental Protection Agency's Research Grant (NV-802-0151-18) to FN in collaboration with the Swedish Agency for Marine and Water Management; the Swedish Research Council VR to P.O.J.H. (Grant no. 2015-03717); the Marie Sklodowska-Curie Individual Fellowship grant to UM (Grant no. 656385). The authors acknowledge support from the National Genomics Infrastructure in Stockholm funded by Science for Life Laboratory, the Knut and Alice Wallenberg Foundation and the Swedish Research Council, and SNIC/Uppsala Multidisciplinary Center for Advanced Computational Science for assistance with massively parallel sequencing and access to the UPPMAX computational infrastructure. We thank Simon Creer for giving feedback on the paper, the captain and crew of R/V Skagerak (University of Gothenburg) for support at sea, Lars B. Pedersen at Aarhus University for sensor constructions, and Volker Brüchert for making the gas chromatographer available at the Department of Geological Sciences, Stockholm University. Open access funding provided by Stockholm University.

## Author contributions

E.B. and S.B. drafted the paper together. E.B. conducted molecular laboratory work, bioinformatics, and molecular data analyses. S.B. sampled in the field, conducted sediment microprofiling, analyzed $N_2O$ samples, and conducted chemistry data analyses. O.H. sorted and identified nematodes and gave feedback on the paper. U.M. helped with chemistry data analysis and gave feedback on the paper. P.H. led the sea expedition and gave feedback on the paper. F.J.A.N. coordinated the study and helped draft the paper. The research was designed by S.B., E.B., and F.J.A.N.

## Competing interests

The authors declare no competing interests.
