## [Peer Review File · Communications Biology]

Reviewers' comments:

Reviewer #1 (Remarks to the Author):

This is an interesting study that combine geochemistry, microscopy and transcriptomic to examine the micrometazoans in oxygen dead zone. The paper is well written and the figures are supportive of the text. I have only few suggestions to offer the authors as to how to improve the text.

- 1) Tables 1 and 2 could be converted in heatmaps. Those would be more intuitive and easier to see the differences.
- 2) Line 101-102 where is this supported in the text? I would suggest to expand a bit more about this.

Reviewer #2 (Remarks to the Author):

This work investigates the taxonomic and functional biodiversity of organisms from dead zone sediments by applying an interdisciplinary approach including geochemistry, morphology, and molecular sequencing tools. Results show that the so-called dead zone sediments are not that dead after all as many organisms show signs of active metabolisms.

More details are found below. Here are my general comments:

I find the dataset very valuable and results interesting. It is very hard to obtain data from such habitats and I appreciate the effort.

However, the manuscript is sometimes confusing and it is unclear what the actual goal and taxonomic focus of the research is. For example, the introduction is almost entirely structured on the effect of dissolved oxygen on the metabolism of meiofauna and protists. But:

Meiofauna include organisms smaller than 2 mm whereas invertebrates obtained by this work are at different size;

Only nematodes, rotifers, and a genus of water fleas (*Bosmina*) are considered in the results; none protist or other meiofauna are mentioned. According to the dataset available in the supplemental material, all the flatworms, tardigrades, gastrotrichs, several annelids – arthropods – nemertean - mollusks, all the gnathostomulids, kinorhynch, the priapulid genus *Tubiluchus* (which is a really interesting finding in such habitats – see: <https://www.jstor.org/stable/24842429?seq=1>), and xenacoelomorphs are meiofauna. Plus, water fleas are technically not meiofauna but macrofauna.

Other parameters, such as hydrogen sulfide and nitrous oxide are equally investigated, not only oxygen, however, the introduction barely mentions H₂S and not at all N₂O;

Prokaryotes are also part of the results, and I find it unrelated to the scope of the manuscript, according to the introduction;

The morphological aspect is very weak and results marginally support the genetic part.

The methods include the sequencing of both RNA and DNA but it is unclear in the results how the two datasets are organized.

Moreover, the statistics need to be improved with additional analyses. I will suggest some examples below.

I also disagree with the idea of including terrestrial organisms in the results as, in my opinion, they are contaminations or inaccuracies present in the Silva genetic database.

Also, the reference list is not updated: many other and more recent works about the biodiversity of organisms (either meiofauna, invertebrates in general, protists, or prokaryotes or all of them according to the main focus of the manuscript) in extreme environments should be cited.

In conclusion, I very much like this work and I wish to see it published after some changes and clarifications.

Specific comments are provided below.

ABSTRACT

The manuscript focuses on differences in dissolved oxygen across four sampling sites. However, also H₂S and N₂O have been considered equally in the analyses and should be given more attention throughout the manuscript. For instance, the effect of N₂O should be mentioned in the abstract. Also, DNA has been investigated, not only RNA, according to the methods.

INTRODUCTION

Line 66. All nematodes have elongated bodies. I am not sure how being elongated enhances the respiratory surface area.

Line 88. Total RNA includes also tRNA and miRNA.

Line 91. It is important to include the statistical correlations among values of oxygen, hydrogen sulfide, and N₂O across the investigated sites. This can be easily done with a linear model/ANOVA. Otherwise, it is not convincing, for example, that decreasing oxygen drives increasing sulfide.

Line 93. Here oxidized nitrogen is mentioned for the first time. Please, explain why it is correlated with anoxic environments. Also 'mRNA transcripts attributed to metazoan in DSZ are significantly different in response to oxygen, oxidized nitrogen, and hydrogen sulfide concentrations.' I don't understand how the transcripts have been quantified. Are they in terms of alpha diversity, number of reads?

Line 100. rRNA (18S) shows taxonomic results. How does it connect with the following sentence ('..zooplankton survive by resting stages..')? Molecular data don't tell at what stage the organism is. And the morphological investigation of this work (more comments below) is not enough to support this statement.

Line 102. Please, cite some works that support that an active COX is related to the possible oxygen sensor. Otherwise, this molecular mechanism needs to be supported by experimental lab work.

Line 108. I feel the authors jump into conclusions too quickly as there is no evidence in this research about the recolonization of nematodes.

The manuscript should cite many recent works focused on meiofauna in extreme environments. See, for example, these reviews and references included:

<https://link.springer.com/article/10.1007/s12526-017-0815-z>

<https://link.springer.com/article/10.1007%2Fs12526-015-0359-z>

RESULTS

Line 114. What's CTD? It is not explained in the methods either. I google it and this is what I found. <https://www.mbari.org/ctd-rosette/> . It seems no to be related to oxygen but Conductivity, Temperature, and Depth.

Line 148. The prokaryotic investigation is out of the scope of this research. The supplementary files about prokaryotes are essentially another paper and it just confuses the reader. There is no need to validate the measured parameters with prokaryotes, as values have been accurately measured with legitimate instruments. Otherwise include also prokaryotes as a larger category named: prokaryotes and invertebrates. This requires re-writing the entire paper. I'd suggest publishing the prokaryotes results in a separated manuscript.

Figure 2 is very nice but it would be much better to also include the measured values in a table.

Line 153 and the next pages. It is often unclear when rRNA comes from the rRNA 18S genes (i.e., DNA) or RNA. It seems like all the taxonomy described in the results has been based on the transcripts (=active organisms). However, the methods indicate some results should come from the 18S genes. Please, organize the results in 2 paragraphs: active organisms (RNA) vs. non-active organisms (DNA). Also, I am not convinced that active organisms are in cryptobiosis as suggested. This should be tested with lab culture experiments. If results obtained from DNA sequences are not shown, delete this protocol from the methods and keep only the RNA.

Statistical Analyses. The analyses considered are fine but not enough to address the goal. I would add at least an additional PERMANOVA to test the effect of oxygen, H₂S, and N₂O on the community structure. That can be done in R, function Adonis.

<https://www.rdocumentation.org/packages/vegan/versions/2.4-2/topics/adonis>

Moreover, since genetic sequences have been obtained, also analyses focused on the phylogenetic diversity could be included. That would clarify if the capability to survive extreme conditions has a phylogenetic signal (phylogenetic regression; e.g., <https://onlinelibrary.wiley.com/doi/full/10.1002/ece3.4031>). That requires the investigation of sub-samples as it is not recommended to investigate phylogenetic diversity on a very taxonomically diverse dataset. Since the manuscript is mainly focused on nematodes and rotifers, you may just consider these two datasets. Alternatively, instead of phylogenetic regression, values of phylogenetic diversity can be easily obtained in R (after reconstructing phylogenetic trees) and the obtained values analyzed with a linear model/ANOVA. This is a suggestion that can be discarded. However, the previous permanova analyses (Adonis) should be included.

This recent paper includes methods and R scripts of some statistical analyses used in meiofaunal community ecology analyzed with metabarcoding 18S. This includes Adonis function in R and phylogenetic diversity: <https://www.nature.com/articles/s42003-018-0119-2>.

This manuscript shows an example of how to use phylogenetic regression in rotifers (although this is at a population level using only a few genes obtained with Sanger sequencing):

<https://link.springer.com/article/10.1007/s10750-016-2725-7>.

From line 192. Take into consideration that the taxonomy associated with Silva can be sometimes (often) wrong. Geneticists and taxonomists should work more together. Please, revise your dataset with accuracy. For example, Eubosmina is no longer an accepted genus. Eubosmina is a genus being eliminated from the literature in favor of Bosmina.

<http://www.marinespecies.org/aphia.php?p=taxdetails&id=148381>

It is unrealistic to find terrestrial species in the middle of the sea at such depths. Those are contaminations of either the samples or of the sequences deposited in Silva. It is necessary to filter the data according to the taxonomy and used only marine invertebrates before performing the statistical analyses. The dataset should be cleaned a lot as I see also sequences of dogs, chickens, bears, camels, gorillas, etc. This is proof that Silva is not fully reliable, as we know already. Unless your assembling is wrong. So, keep only the ones that are credibly present in your deep-sea samples. Also, the supplemental files are not named according to the text and I find hard to figure which one to look at. I am now looking at the one named 3198_0_supp_77617_q0wtw7. I see that other invertebrates, such as flatworms, annelids, mollusks are way more abundant than rotifers. Please, justify the choice of rotifers and nematodes.

Microscopic analyses. It is good to compare genetic results with microscopic observations. Is it possible to look at all the samples and not just station E? Also, no rotifers or other invertebrates have been observed? Just nematodes and eggs? How can you be that sure that such eggs belong to Bosmina? You can be sure only if you hatch them. The morphological aspect is very limited compared to the genetic one.

DISCUSSION

Line 268. Replace 'the latest' with another word, such as up-to-date.

Line 333. *Bosmina* must be written in italic. Check it also throughout the text.

Line 356-370. I disagree with this paragraph. I would not justify the presence of freshwater species if not with either (i) contamination of the samples, (ii) contamination of sequence in the database, or (iii) wrong sequence assembling/taxonomic assignment. See above. It is a common practice to filter the sequences and keep only the ones that make sense. Of course, this should be mentioned in the methods. In this case, only marine invertebrates should be considered. That requires a long and tedious work throughout all the taxa but necessary to make sure that results are not misleading. Protists are not discussed, so, they should be taken off the manuscript entirely or discussed.

MATERIAL AND METHODS

Page 20. What is the length of RNA sequences that match the assembling? It is confusing to me the distinction between RNA and DNA sequencing. The molecular section should be thoroughly organized.

REFERENCES

Once the focus of the manuscript is clearer (meiofauna? Invertebrates? Protists? Prokaryotes? All of them?) more references about the adaptation of such organisms in extreme/anoxic environments should be included.

Reviewer #1 (Remarks to the Author):

This is an interesting study that combine geochemistry, microscopy and transcriptomic to examine the micrometazoans in oxygen dead zone. The paper is well written and the figures are supportive of the text. I have only few suggestions to offer the authors as to how to improve the text.

Reply: We thank the reviewer for the positive comments and have revised the manuscript according to the reviewer's suggestions.

1) Tables 1 and 2 could be converted in heatmaps. Those would be more intuitive and easier to see the differences.

Reply: We have converted Table 1 and 2 into heatmaps (now figure 5 and 6, respectively).

2) Line 101-102 where is this supported in the text? I would suggest to expand a bit more about this.

Reply: We have expanded on this in the introduction: "Many pelagic zooplankton organisms have benthic stages and can survive hypoxic/anoxic conditions in the form of resting eggs^{8,9}, such eggs have been shown to hatch once oxygen returns¹⁰." (lines 63-65)

And clarified this at the end of the introduction: "Our results indicate that zooplankton are present as resting stages in DZS, and the mRNA data suggest that these organisms use the enzyme cytochrome c oxidase as an oxygen sensor (which has previously been shown in e.g. yeast²⁸)." (lines 104-106)

This is discussed in more detail in the discussion: "Rotifer egg banks (and other zooplankton) have also been previously observed in Baltic Sea anoxic sediments and to hatch upon oxygenation¹⁰." (lines 322-323)

Reviewer #2 (Remarks to the Author):

This work investigates the taxonomic and functional biodiversity of organisms from dead zone sediments by applying an interdisciplinary approach including geochemistry, morphology, and molecular sequencing tools. Results show that the so-called dead zone sediments are not that dead after all as many organisms show signs of active metabolisms.

More details are found below. Here are my general comments:

I find the dataset very valuable and results interesting. It is very hard to obtain data from such habitats and I appreciate the effort.

However, the manuscript is sometimes confusing and it is unclear what the actual goal and

taxonomic focus of the research is. For example, the introduction is almost entirely structured on the effect of dissolved oxygen on the metabolism of meiofauna and protists.

But: Meiofauna include organisms smaller than 2 mm whereas invertebrates obtained by this work are at different size;

Only nematodes, rotifers, and a genus of water fleas (*Bosmina*) are considered in the results; none protist or other meiofauna are mentioned. According to the dataset available in the supplemental material, all the flatworms, tardigrades, gastrotrichs, several annelids – arthropods – nemertean - mollusks, all the gnathostomulids, kinorhynchans, the priapulid genus *Tubiluchus* (which is a really interesting finding in such habitats – see: <https://www.jstor.org/stable/24842429?seq=1>), and xenacoelomorphs are meiofauna. Plus, water fleas are technically not meiofauna but macrofauna.

Other parameters, such as hydrogen sulfide and nitrous oxide are equally investigated, not only oxygen, however, the introduction barely mentions H₂S and not at all N₂O;

Prokaryotes are also part of the results, and I find it unrelated to the scope of the manuscript, according to the introduction;

The morphological aspect is very weak and results marginally support the genetic part.

The methods include the sequencing of both RNA and DNA but it is unclear in the results how the two datasets are organized.

Moreover, the statistics need to be improved with additional analyses. I will suggest some examples below.

I also disagree with the idea of including terrestrial organisms in the results as, in my opinion, they are contaminations or inaccuracies present in the Silva genetic database.

Also, the reference list is not updated: many other and more recent works about the biodiversity of organisms (either meiofauna, invertebrates in general, protists, or prokaryotes or all of them according to the main focus of the manuscript) in extreme environments should be cited.

In conclusion, I very much like this work and I wish to see it published after some changes and clarifications.

Reply: We thank the reviewer for the positive comments and interest in the manuscript. We revised the manuscript whenever possible according to the reviewer's suggestions (see specific comments below).

Specific comments are provided below.

ABSTRACT

The manuscript focuses on differences in dissolved oxygen across four sampling sites. However, also H₂S and N₂O have been considered equally in the analyses and should be given more attention throughout the manuscript. For instance, the effect of N₂O should be mentioned in the abstract.

Reply: We have now included N₂O in the abstract: “Here we combined geochemistry, microscopy and RNA-seq for estimating taxonomy and functionality of active micrometazoans along an oxygen gradient, with differences in nitrous oxide and hydrogen sulfide concentrations, in the largest dead zone in the world.” (lines 29-32)

and: “We show that the diversity and community structure of metazoan in DZS are different between low oxygen areas, which may be primarily due to porewater concentrations of hydrogen sulfide and its potential to be oxidized with oxygen or nitrate.” (lines 35-38)

We have also added a few sentences in the introduction to introduce N₂O: “Under anoxic conditions and when nitrate is present, such bacteria are known to couple sulfide oxidation with nitrate reduction^{25, 26} and this process may yield oxidized nitrogen compounds such as nitrous oxide (N₂O)^{25, 26}. N₂O has therefore been shown to be a good indicator of potential nitrate reduction at the oxic-anoxic interface of the Baltic Sea dead zone²⁷.” (lines 84-87)

Also, DNA has been investigated, not only RNA, according to the methods.

Reply: We have removed the mentioning of DNA from the abstract, as we only used RNA to investigate alive animals.

INTRODUCTION

Line 66. All nematodes have elongated bodies. I am not sure how being elongated enhances the respiratory surface area.

Reply: This sentence has been rewritten, it now reads: “Nematodes are among the most abundant animals in these regions^{12, 13, 14} and have evolved strategies to cope with low oxygen conditions^{15, 16}.” (lines 66-68)

Line 88. Total RNA includes also tRNA and miRNA.

Reply: We have clarified this sentence to better reflect the study, it now reads: “To our knowledge, there are no studies using total RNA sequencing to analyze both rRNA plus mRNA to investigate dead zone animals.” (lines 90-91)

Line 91. It is important to include the statistical correlations among values of oxygen, hydrogen sulfide, and N₂O across the investigated sites. This can be easily done with a linear model/ANOVA. Otherwise, it is not convincing, for example, that decreasing oxygen drives increasing sulfide.

Reply: We have now done a correlations analysis based on all O₂, H₂S, and N₂O values from the measured microprofiles (full results table in Supplementary Table 1).

This has been added to the results: “O₂ correlated negatively with H₂S ($\rho = -0.76$, $P < 0.001$) and positively with N₂O ($\rho = 0.85$, $P < 0.001$) in the measured sediment cores (tested for the whole dataset from all stations, Spearman correlations; Supplementary Table 1).” (lines 130-133)

We have also updated the aim to be clearer about what we hypothesized (rather than writing about decreasing oxygen and increasing sulfide considering we do not have data over time): “(1) low oxygen and high sulfide concentrations reduce metazoan diversity and alter community structure” (lines 94-95)

Line 93. Here oxidized nitrogen is mentioned for the first time. Please, explain why it is correlated with anoxic environments. Also ‘mRNA transcripts attributed to metazoan in DSZ are significantly different in response to oxygen, oxidized nitrogen, and hydrogen sulfide concentrations.’ I don’t understand how the transcripts have been quantified. Are they in terms of alpha diversity, number of reads?

Reply: We have also added a few sentences in the introduction to introduce N₂O:”Under anoxic conditions and when nitrate is present, such bacteria are known to couple sulfide oxidation with nitrate reduction^{25, 26} and this process may yield oxidized nitrogen compounds such as nitrous oxide (N₂O)^{25, 26}. N₂O has therefore been shown to be a good indicator of potential nitrate reduction at the oxic-anoxic interface of the Baltic Sea dead zone²⁷.” (lines 84-87)

We have clarified this sentence in the aims, it now reads: “(2) mRNA transcripts translating for metazoan proteins in DZS are significantly different (in amount and function) in response to oxygen, nitrous oxide, and hydrogen sulfide concentrations.” (lines 95-97)

Line 100. rRNA (18S) shows taxonomic results. How does it connect with the following sentence (..zooplankton survive by resting stages...)? Molecular data don’t tell at what stage the organism is. And the morphological investigation of this work (more comments below) is not enough to support this statement.

Reply: Here we summarize our results at the end of the Introduction. Previous studies have found that zooplankton survive by resting stages in anoxic sediment. We now mention this in the introduction, discuss this and cite this literature in the discussion of the manuscript. We have clarified this sentence at the end of the Introduction as well.

We have expanded on the presence of zooplankton in anoxic sediment in the introduction: “Many pelagic zooplankton organisms have benthic stages and can survive hypoxic/anoxic conditions in the form of resting eggs^{8, 9}, such eggs have been shown to hatch once oxygen returns¹⁰.” (lines 63-65)

And clarified our results better at the end of the introduction: “Our results indicate that zooplankton are present as resting stages in DZS, and the mRNA data suggest that these organisms use the enzyme cytochrome c oxidase as an oxygen sensor (which has previously been shown in e.g. yeast²⁸).” (lines 104-106)

This is discussed in more detail in the discussion: “Rotifer egg banks (and other zooplankton) have also been previously observed in Baltic Sea anoxic sediments and to hatch upon oxygenation¹⁰.” (lines 322-323)

Line 102. Please, cite some works that support that an active COX is related to the possible oxygen sensor. Otherwise, this molecular mechanism needs to be supported by experimental lab work.

Reply: We cite previous studies in the discussion: “In the hypoxic/anoxic sediments predominant portion of RNA transcripts affiliated with pelagic taxa like Bosmina (formerly Eubosmina) and Rotifera were attributed to COX subunit I. This protein can be used as an oxygen sensor as seen for mammalian tissue cells³⁶ and yeast²².” (line 310-312)

We have now also clarified this at the end of the Introduction: “Our results indicate that zooplankton are present as resting stages in DZS, and the mRNA data suggest that these organisms use the enzyme cytochrome c oxidase as an oxygen sensor (which has previously been shown in e.g. yeast²⁸).” (lines 104-106)

Line 108. I feel the authors jump into conclusions too quickly as there is no evidence in this research about the recolonization of nematodes.

Reply: We have removed these sentences regarding recolonization of nematodes at the end of the introduction.

The manuscript should cite many recent works focused on meiofauna in extreme environments. See, for example, these reviews and references included:

<https://link.springer.com/article/10.1007/s12526-017-0815-z>

<https://link.springer.com/article/10.1007%2Fs12526-015-0359-z>

Reply: We now cite these two references (and others) in the introduction.

RESULTS

Line 114. What’s CTD? It is not explained in the methods either. I google it and this is what I found. <https://www.mbari.org/ctd-rosette/> . It seems no to be related to oxygen but Conductivity, Temperature, and Depth.

Reply: We have clarified that the sampler was equipped with O₂ sensors and added details of the CTD system in the methods: “Water column oxygen profiles were measured by means of a CTD-rosette system (SBE 911plus, SeaBird Electronics, USA) equipped with O₂ sensors (SBE 43 Dissolved O₂ Sensor, SeaBird Electronics, USA).” (lines 364-366)

We also decided to remove the mentioning of CTD at the beginning of the results as this fit better in the methods section.

Line 148. The prokaryotic investigation is out of the scope of this research. The supplementary files about prokaryotes are essentially another paper and it just confuses the reader. There is no need to validate the measured parameters with prokaryotes, as values have been accurately measured with legitimate instruments. Otherwise include also prokaryotes as a larger category named: prokaryotes and invertebrates. This requires re-writing the entire paper. I'd suggest publishing the prokaryotes results in a separated manuscript.

Reply: We have removed sections that were related to the prokaryotic data analysis in the manuscript, methods, and supplemental files. We agree with the reviewer that this has helped with focusing the manuscript on its scope.

Figure 2 is very nice but it would be much better to also include the measured values in a table.

Reply: We have added sediment microprofile data into a table (Table 1).

Line 153 and the next pages. It is often unclear when rRNA comes from the rRNA 18S genes (i.e., DNA) or RNA. It seems like all the taxonomy described in the results has been based on the transcripts (=active organisms). However, the methods indicate some results should come from the 18S genes. Please, organize the results in 2 paragraphs: active organisms (RNA) vs. non-active organisms (DNA). Also, I am not convinced that active organisms are in cryptobiosis as suggested. This should be tested with lab culture experiments. If results obtained from DNA sequences are not shown, delete this protocol from the methods and keep only the RNA.

Reply: The reviewer is correct that only RNA data was used to investigate the eukaryotes, with a focus on Metazoan, as we wanted to focus on live animals. We have removed DNA methods from the manuscript and supplemental files.

We also decided to follow the reviewer's recommendation to tone down the statements on cryptobiosis in the discussion: "Considering the lower amount of sequences and the absence of essential enzymes for transcription and translation at stations D and F, it is possible that nematode communities at these stations consisted of low abundant taxa adapted or trying to survive in these extreme conditions." (lines 280-283)

And conclusions: "Nematodes survive in specialized niches such as sulfide oxidation zones, or are in low abundance (potentially with a downregulated metabolism) in anoxic and sulfidic sediments." (lines 336-338)

And at the end of the introduction: "Additionally, nematodes can persist in anoxic and sulfidic sediments in niches like sulfide oxidation zones, or in low abundance (potentially with a downregulated metabolism)." (lines 106-108)

Statistical Analyses. The analyses considered are fine but not enough to address the goal. I would add at least an additional PERMANOVA to test the effect of oxygen, H₂S, and N₂O on the community structure. That can be done in R, function Adonis.

<https://www.rdocumentation.org/packages/vegan/versions/2.4-2/topics/adonis>

Reply: While we understand the reviewers point, this analysis is not possible in our case because the sediment microprofiles and community composition were measured in different sediment cores. As such, we are unable to directly link the variation of our abiotic variables to the changes in community composition as they were measured in different sampling units. We were forced to follow this sampling strategy due to the specific logistical challenges of our study. Sediment microprofiling is a time-consuming procedure (especially with several replicate profiles in the same core as in this study) and it was therefore not an option to slice such cores for RNA extraction as mRNA has a very short lifetime. To avoid losing mRNA signal and maximize sampling sediment for RNA as close to in situ conditions as possible, sediment for RNA extraction was therefore collected from other cores as soon as they reached the surface.

However, as the reviewer has seen in the manuscript we have conducted an NMDS (with PERMANOVA adonis analysis) based on the beta diversity of the eukaryotic community (Figure 3). This figure shows that station A (oxic and no sulfide) clusters differently from the other hypoxic-anoxic stations that all had very low-to-none oxygen but high concentrations of sulfide. This analysis also shows that that station E clustered different from D and F due to the higher concentration of N₂O.

We have nevertheless tried to alleviate the reviewer concerns by clarifying these results in the manuscript: “NMDS analysis of eukaryotic beta diversity showed that the stations formed different clusters, especially station A (O₂ rich and no H₂S) compared to the hypoxic-anoxic stations that all had higher concentrations of sulfide, when tested for presence/absence and the relative abundance (PERMANOVA, Sørensen index and Bray-Curtis dissimilarity, $F = 13.4$ and $F = 43.1$, respectively, $P < 0.01$ for both tests; Sørensen Fig. 3b, and Bray-Curtis in Supplementary Fig. 1). In the same analysis, station E that had the highest concentration of N₂O clustered differently when compared to the other hypoxic-anoxic stations D and F.” (lines 156-163)

Moreover, since genetic sequences have been obtained, also analyses focused on the phylogenetic diversity could be included. That would clarify if the capability to survive extreme conditions has a phylogenetic signal (phylogenetic regression; e.g., <https://onlinelibrary.wiley.com/doi/full/10.1002/ece3.4031>). That requires the investigation of sub-samples as it is not recommended to investigate phylogenetic diversity on a very taxonomically diverse dataset. Since the manuscript is mainly focused on nematodes and rotifers, you may just consider these two datasets. Alternatively, instead of phylogenetic regression, values of phylogenetic diversity can be easily obtained in R (after reconstructing phylogenetic trees) and the obtained values analyzed with a linear model/ANOVA. This is a suggestion that can be discarded. However, the previous permanova analyses (Adonis) should be included.

This recent paper includes methods and R scripts of some statistical analyses used in meiofaunal community ecology analyzed with metabarcoding 18S. This includes Adonis function in R and phylogenetic diversity: <https://www.nature.com/articles/s42003-018-0119-2>.

This manuscript shows an example of how to use phylogenetic regression in rotifers (although this is at a population level using only a few genes obtained with Sanger sequencing): <https://link.springer.com/article/10.1007/s10750-016-2725-7>.

Reply: As mentioned by the reviewer we decided to discard the phylogenetic analysis. Such data has to be conducted on DNA and would therefore not reflect alive metazoan which our study focuses on. As such the phylogenetic signal of alive metazoan would most probably be confounded by the large amount of DNA from dead organisms present in the sediment.

While this suggestion by the reviewer would be interesting to explore this in our opinion falls outside the scope of our aims. In addition, as indicated by the literature suggested by the reviewer this kind of analyses would be much better if based on barcoding or metabarcoding data as all the 18S rRNA fragments spans the same region and are comparable. To perform such analysis with our data based on high-throughput shotgun sequencing the bioinformatic analysis is quite complex. This would include the need to construct assembled DNA data from each station (as we are comparing phylogenetic differences of similar taxa between stations). Contigs determined as 18S rRNA genes would need to be extracted and classified to taxonomy. Then arthropod and nematode data would need to be filtered to only include contigs spanning similar regions on the 18S rRNA gene. This is a very time and resource intensive task which we feel would add limited complementary information to our conclusions.

From line 192. Take into consideration that the taxonomy associated with Silva can be sometimes (often) wrong. Geneticists and taxonomists should work more together. Please, revise your dataset with accuracy. For example, Eubosmina is no longer an accepted genus. Eubosmina is a genus being eliminated from the literature in favor of Bosmina.

<http://www.marinespecies.org/aphia.php?p=taxdetails&id=148381>

Reply: We were able to find rather recent literature still separating the labels Bosmina and Eubosmina. E.g. <https://link.springer.com/article/10.1007/s00300-018-2282-9> <https://academic.oup.com/plankt/article/41/3/273/5423651> <https://aslopubs.onlinelibrary.wiley.com/doi/abs/10.4319/lo.2011.56.2.0440> likely due to the database containing these classifications separately.

We prefer to show the results as it was determined by the database used (as done in the example articles above). We therefore decided to instead indicate in the manuscript that Bosmina was formerly labelled as Eubosmina). (lines 311)

It is unrealistic to find terrestrial species in the middle of the sea at such depths. Those are contaminations of either the samples or of the sequences deposited in Silva. It is necessary to filter the data according to the taxonomy and used only marine invertebrates before performing

the statistical analyses. The dataset should be cleaned a lot as I see also sequences of dogs, chickens, bears, camels, gorillas, etc. This is proof that Silva is not fully reliable, as we know already. Unless your assembling is wrong. So, keep only the ones that are credibly present in your deep-sea samples.

Reply: We have updated Figure 4 with the terrestrial taxa grouped into “unclassified”. We have also added a new sentence into methods to clarify this: “When visualizing lower taxonomic levels (Fig. 4b and 4c) freshwater and terrestrial taxa that were likely derived from database errors (Arthropoda: Teloganopsis, Stenchaetothrips, Metatriconiscoides; and Nematoda: Fictor and Strongyloides) were included in the group “unclassified”. A full list of all classifications is available in Supplementary Data 2.” (lines 451-455)

We decided to keep the supplemental files as they are, and show the results as there were retrieved from the databases. Regarding hits for mouse or human this does not influence our results or conclusions for this study when comparing nematoda and arthropods between sites. Also, Vertebrata had a very low presence in the dataset as indicated by Fig. 4a.

Also, the supplemental files are not named according to the text and I find hard to figure which one to look at. I am now looking at the one named 3198_0_supp_77617_q0wtw7. I see that other invertebrates, such as flatworms, annelids, mollusks are way more abundant than rotifers. Please, justify the choice of rotifers and nematodes.

Reply: We are sorry for the confusing names of the supplemental files. It seems that the submission system renamed the files once the manuscript was submitted. As noted by the reviewer only the RNA data was used to investigate the eukaryotes, with a focus on alive Metazoan. We have therefore removed DNA data from the supplemental files, this should help to make the data easier to digest. The reviewer is also correct that many flatworms, annelids, mollusks etc., have reads assigned, but this is for site A (i.e. the oxic station). Fig. 4A and Supplementary Figure 2 gives a good overview of how classified reads were distributed between taxa and stations after taxonomic classification.

Microscopic analyses. It is good to compare genetic results with microscopic observations. Is it possible to look at all the samples and not just station E? Also, no rotifers or other invertebrates have been observed? Just nematodes and eggs? How can you be that sure that such eggs belong to *Bosmina*? You can be sure only if you hatch them. The morphological aspect is very limited compared to the genetic one.

Reply: We only investigated the station E samples microscopely because of labor intensive work (i.e. samples were not sieved due to being frozen beforehand, and we were also unsure of the size of potential animals). We therefore decided to focus on the station with the highest chance of finding nematodes based on the RNA data. Also, the morphological study was done to support and verify the results of the molecular work, and are not meant to be comprehensive.

We did not detect any rotifera. If there are rotifera might be in the form of resting eggs picked up in the RNA data. In the study we do not rule out rotifera, as the RNA indicates that they were there. Potentially there are eggs there but we were unable to identify them. Regarding other

animals such as mollusca, etc, we did not find other animals in the station E samples, and the RNA data indicated that such organisms would mainly be present in the oxic station (as shown in Fig 4).

According to the known morphology of *Bosmina* and our microscopy plus RNA results we think it's very likely that the samples contained *Bosmina*. We have more photos from the microscope samples (we couldn't include them all in the manuscript). And here are some more of the zooplankton. We have also included some of these photos now as Supplementary Figure 3. Here the eggs can be seen inside the carapaci of what looks very similar to *Bosmina*.

To add uncertainty to the microscope zooplankton results in the manuscript we decided to change the word “*Bosminidae*” to “*Bosminidae-like*”.

DISCUSSION

Line 268. Replace ‘the latest’ with another word, such as up-to-date.

Reply: We changed to up-to-date as suggested by the reviewer.

Line 333. *Bosmina* must be written in italic. Check it also throughout the text.

Reply: This have been fixed here and throughout the manuscript.

Line 356-370. I disagree with this paragraph. I would not justify the presence of freshwater

species if not with either (i) contamination of the samples, (ii) contamination of sequence in the database, or (iii) wrong sequence assembling/taxonomic assignment. See above. It is a common practice to filter the sequences and keep only the ones that make sense. Of course, this should be mentioned in the methods. In this case, only marine invertebrates should be considered. That requires a long and tedious work throughout all the taxa but necessary to make sure that results are not misleading.

Reply: We decided to keep the supplemental files as they are, and show the results as there were retrieved from the databases. This might be useful in the future if and when the databases are corrected. Considering that these freshwater and terrestrial taxa might be marine/brackish but erroneously classified in the database we decided to keep them but group them as unclassified when shown in the manuscript (Fig. 4b).

We have updated Figure 4 with these changes and we have also added a new sentence into methods to clarify this: “When visualizing lower taxonomic levels (Fig. 4b and 4c) freshwater and terrestrial taxa that were likely derived from database errors (Arthropoda: Teloganopsis, Stenchaetothrips, Metatriconiscoides; and Nematoda: Fictor and Strongyloides) were included in the group “unclassified”. A full list of all classifications is available in Supplementary Data 2.” (lines 451-455)

We have also removed the discussion regarding the terrestrial taxa.

Protists are not discussed, so, they should be taken off the manuscript entirely or discussed.

Reply: We have removed the paragraph regarding protists in the introduction, results and discussion.

MATERIAL AND METHODS

Page 20. What is the length of RNA sequences that match the assembling? It is confusing to me the distinction between RNA and DNA sequencing. The molecular session should be thoroughly organized.

Reply: We have removed sections that were related to the prokaryotic data analysis in the manuscript, methods (that includes the metagenome DNA assembly), and supplemental files. This has made the methods to have a better flow and easier to follow.

Even though the DNA metagenome assembly was conducted with the prokaryotes in mind, i.e. using translation table 11 for prokaryotic gene expression (default setting on the most typical gene prediction tools), we were able to detect mapped RNA transcripts for e.g. nematodes as well and this was previously included as supplementary information only.

However, we got good results detecting animals in the DZS with the MEGAN+DIAMOND approach that is used to classify merged paired-end non-rRNA reads (following the published SAMSA2 pipeline). The manuscript therefore focused on these findings instead. Such an

approach would also help to detect rare animals and their affiliated proteins that are not assembled or lost during mapping of RNA reads.

We have now clarified in the manuscript what bioinformatic approach we used in the methods at the start of the protein classification section. We agree that the changes suggested by the reviewers have made the bioinformatic methods more organized and easier to follow.

REFERENCES

Once the focus of the manuscript is clearer (meiofauna? Invertebrates? Protists? Prokaryotes? All of them?) more references about the adaptation of such organisms in extreme/anoxic environments should be included.

Reply: We have streamlined the introduction, results, and discussion to focus on the main points in the manuscript. Prokaryotes have been removed from the results and methods (as suggested by the reviewer). We have also removed protists in the introduction and results (also suggested by the reviewer). The manuscript is now therefore focused on micrometazoan.

One paragraph in the introduction has been rewritten to better reflect the focus of the study: “Many pelagic zooplankton organisms have benthic stages and can survive hypoxic/anoxic conditions in the form of resting eggs^{8,9}, such eggs have been shown to hatch once oxygen returns¹⁰. However, some eukaryotic organisms are adapted to live in anoxia, which may be due to presence of copious organic matter and low predation pressure^{6,11}. Nematodes are among the most abundant animals in these regions^{12,13,14} and have evolved strategies to cope with low oxygen conditions^{15,16}.” (lines 63-68)

REVIEWERS' COMMENTS:

Reviewer #2 (Remarks to the Author):

I very much appreciate how the authors took comments very seriously and replied to them in great detail.

I hope to see this very interesting paper published soon on Communication Biology.
Congratulations on this very fine work!